# A fast *Myosin* super enhancer dictates muscle fiber phenotype through competitive interactions with *Myosin* genes

Matthieu Dos Santos[1], Stéphanie Backer[1], Frédéric Auradé[2,3], Matthew Man-Kin Wong [4], Maud Wurmser[1], Rémi Pierre[1], Francina Langa [5], Marcio Do Cruzeiro[1], Alain Schmitt[1], Jean-Paul Concordet [6], Athanassia Sotiropoulos[1], F. Jeffrey Dilworth [4], Daan Noordermeer [7], Frédéric Relaix [2], Iori Sakakibara[1,8✉] & Pascal Maire [1✉]

The contractile properties of adult myofibers are shaped by their Myosin heavy chain isoform content. Here, we identify by snATAC-seq a 42 kb super-enhancer at the locus regrouping the fast *Myosin* genes. By 4C-seq we show that active fast *Myosin* promoters interact with this super-enhancer by DNA looping, leading to the activation of a single promoter per nucleus. A rainbow mouse transgenic model of the locus including the super-enhancer recapitulates the endogenous spatio-temporal expression of adult fast *Myosin* genes. In situ deletion of the super-enhancer by CRISPR/Cas9 editing demonstrates its major role in the control of associated fast *Myosin* genes, and deletion of two fast *Myosin* genes at the locus reveals an active competition of the promoters for the shared super-enhancer. Last, by disrupting the organization of fast *Myosin*, we uncover positional heterogeneity within limb skeletal muscles that may underlie selective muscle susceptibility to damage in certain myopathies.

[1] Université de Paris, Institut Cochin, INSERM, CNRS, 75014 Paris, France. [2] Univ Paris Est Creteil, INSERM, EnvA, EFS, AP-HP, IMRB, 94010 Creteil, France. [3] Sorbonne Université, INSERM U974, Center for Research in Myology, 75013 Paris, France. [4] Regenerative Medicine Program. Ottawa Hospital Research Institute, Ottawa, Canada. [5] Institut Pasteur, Paris, France. [6] MNHN, Paris, France. [7] Université Paris-Saclay, CEA, CNRS, Institute for Integrative Biology of the Cell (I2BC), Gif-sur-Yvette, France. [8] Institute of Medical Nutrition, Tokushima University Graduate School, Tokushima 770-8503, Japan. ✉email: iori.f.sakakibara@gmail.com; pascal.maire@inserm.fr

Skeletal muscles constitute the most abundant organ in an adult human, ~40% of its total body mass. Most skeletal muscles are composed of a mixture of myofibers with distinct contractile, metabolic, resistance to fatigue properties, as well as differential vulnerability in pathophysiological situations[1]. These different myofibers can be classified as slow or fast subtypes that selectively express genes responsible for their specific properties[2–4]. The most widely used classification of myofibers types is based on their Myosin heavy chain (MYH) expression profile[4–7]. MYH, one of the most abundant proteins present in adult myofibers, is a major determinant of myofiber speed of contraction. Each of the mammalian MYH isoform is coded by a specific gene and adult slow-type myofibers express *Myh7* (also known as *MyHCI*, *β* or *slow*), adult fast-type myofibers express *Myh2* (*MyHCIIA*), *Myh1* (*MyHCIIX*), *Myh4* (*MyHCIIB*), or *Myh13* (*MyHCeo*). During embryonic development *Myh7* and two specific fast *Myh* (f*Myh*) genes, *Myh3* (*MyHCemb*), and *Myh8* (*MyHCperi*) are expressed[8].

The f*Myh* genes (*Myh3, Myh2, Myh1, Myh4, Myh8,* and *Myh13*) are organized as a cluster within a 350 kb region on mouse chromosome 11[9]. The adult fast *Myh2, Myh1* and *Myh4* genes are expressed at a low-level during embryogenesis and start to be expressed at a much higher level after birth[8,10–12]. The mechanisms controlling the robust coordinated expression of f*Myh* genes in the hundreds nuclei of a myofiber are not understood. Special regulatory elements called super enhancers (SE) have been shown to control high expression levels for cell lineage identity genes. These SE are composed of multiple enhancer elements spanning 10–50 kb of DNA and allowing efficient expression of associated genes[13–18]. As identity genes expressed at high levels in specific fast myofiber subtypes, f*Myh* genes are good candidates to be controlled by a SE in the skeletal muscle lineage. The clustered organization and strict temporal regulation of the f*Myh* locus shows similarities with that of the human *β-globin* locus[19]. At the *β-globin* locus a common regulatory sequence called locus control region (LCR) interacts dynamically with the different promoters within the locus to activate a single *Globin* isoform in erythroid cells[20–22]. We hypothesized that a LCR/SE at the f*Myh* locus may coordinate the expression of selective f*Myh* genes in adult myofibers to finely control their identity.

To characterize the cis-regulatory elements required for the complex regulation of the specific f*Myh* genes we performed snATAC-seq and 4C-seq experiments with adult skeletal muscles and identified a 42-kb opened chromatin region interacting in an exclusive manner with the activated f*Myh* promoter at the locus through 3D chromatin looping as revealed by 4C-seq experiments. A mouse rainbow transgenic line including this SE recapitulates the spatio-temporal expression of endogenous *Myh2, Myh1,* and *Myh4* genes. We further show by CRISPR/Cas9 editing that in situ deletion of this 42 kb SE region prevents expression of fetal *Myh8* and adult f*Myh* genes at the locus leading to fetal myofibers devoid of sarcomeres, unable to contract and precluding breathing at birth. We also tested the hypothesis of promoter competition for the shared SE and show that absence of *Myh1* and *Myh4* leads to increased expression of *Myh2, Myh8,* or *Myh13* in specific subregions of limb muscles. Altogether our studies demonstrate that the f*Myh* SE is responsible for the non-stochastic robust coordinated f*Myh* gene expression in the hundreds of body myonuclei present in adult myofibers. Analysis of the phenotype of all forelimbs and hindlimbs muscles in genetic perturbations within the f*Myh* locus reveals different categories of muscle susceptibility reminiscent of the selective muscle vulnerability observed in different neuromuscular diseases.

## Results

### Identification of a super enhancer acting as a locus control region in the f*Myh* locus.
The majority of adult myofibers express a single *Myh* gene among the subfamily of fast *Myh4, Myh1, Myh2,* or slow *Myh7* genes. Fast muscles like the quadriceps are composed of myofibers expressing predominantly *Myh4* or *Myh1* genes while slow muscles like the soleus are composed of myofibers expressing predominantly *Myh7* or *Myh2* genes (Fig. 1A, B). To identify the regulatory elements controlling the expression of f*Myh* genes, we performed snATAC-seq experiments with nuclei isolated from adult fast quadriceps and slow soleus[10]. Myonuclei were classified based on the chromatin accessibility in the promoter and gene body of *Myh* genes (Figs. 1C, D and S1A). In f*Myh* myonuclei (*Myh2, Myh1, and Myh4*), we observed 7 chromatin accessibility peaks in an intergenic region between *Myh3* and *Myh2* (Fig. S1B). This chromatin region is "closed" in nuclei that do not express f*Myh* genes like slow *Myh7* myonuclei and Fibro Adipogenic Progenitors (FAPs) nuclei where no snATAC-seq peak is detected (Figs. 1D and S1A). These chromatin accessibility peaks cover the *Linc-Myh* gene[23], and end 25 kb upstream of *Myh2* promoter (Fig. S1B). Because of its large size of 42 kb, this element could correspond to a conserved super enhancer (SE) controlling the f*Myh* genes of the locus in mammals (Fig. S1B).

SE was first identified by Chip-seq by their higher level of transcription coactivators and active histone marks accumulation (H3K4me2 and H3K27ac) than conventional enhancers, and by their larger size compared with classic enhancers[24]. They regulate the expression of highly transcribed genes specifying cell identity. To test if the element that we identified corresponds to a SE, we compared snATAC-seq results with available ChIP-seq performed in adult skeletal muscle against H3K4me2 and H3K27ac histone marks[25], and against histone methyl transferase MLL4 (KMT2D, a MEF2 transcriptional cofactor)[26] (Fig. S1B). We observed specific enrichment of these two active histone marks in the same snATAC-seq peaks of chromatin accessibility in the intergenic region between *Myh3* and *Myh2* in quadriceps and soleus (Figs. 1E and S1A[25]). To determine if this sequence is a SE, we classified the slow and fast specific muscle active enhancers according to the enrichment in H3K27ac histone marks. The 42-kb regulatory region of the f*Myh* locus shows a strong enrichment in H3K27ac marks compared to the other enhancers (Fig. 1F). This showed that this 42 kb intergenic region between *Myh3* and *Myh2* possesses the characteristics of a SE[15] that could control the expression of adjacent f*Myh* genes in the fast quadriceps but also in the slow soleus where around 50% of myofibers express *Myh2* or *Myh1* (Figs. 1G and S1A). Based on these criteria, we named this regulatory element f*Myh*-SE.

One of the first SE discovered was the locus control region (LCR) of the *β-globin* locus[18,27]. Like the f*Myh* locus, the human *β-globin* locus contains a cluster of *globin* gene isoforms expressed sequentially during embryonic, fetal, and adult erythropoiesis[28]. The LCR of the *β-globin* locus forms dynamical and specific chromatin loops with the promoter of the gene transcribed at the locus. The similarities between clustered organization and temporal expression at the *β-globin* and f*Myh* loci suggested that the f*Myh*-SE could act by chromatin looping. To verify this, we performed 4C-seq by purifying nuclei from fast quadriceps and slow soleus. We designed specific primers to quantify the DNA regions interacting with *Myh4* and *Myh2* promoters when these genes are expressed or not. We observed that the *Myh4* promoter interacted significantly more with the f*Myh*-SE in the quadriceps where the corresponding gene is more transcribed than in the soleus (Figs. 1H and S2A). On the contrary, we observed significantly more interactions between the *Myh2* promoter and the f*Myh*-SE in the soleus where the corresponding gene is more transcribed than in the quadriceps. We confirmed these results by quantifying the interactions between the f*Myh*-SE and other DNA regions in fast muscles. We observed strong and specific interactions between the f*Myh*-SE and the *Myh4* promoter in muscles expressing predominantly *Myh4* gene

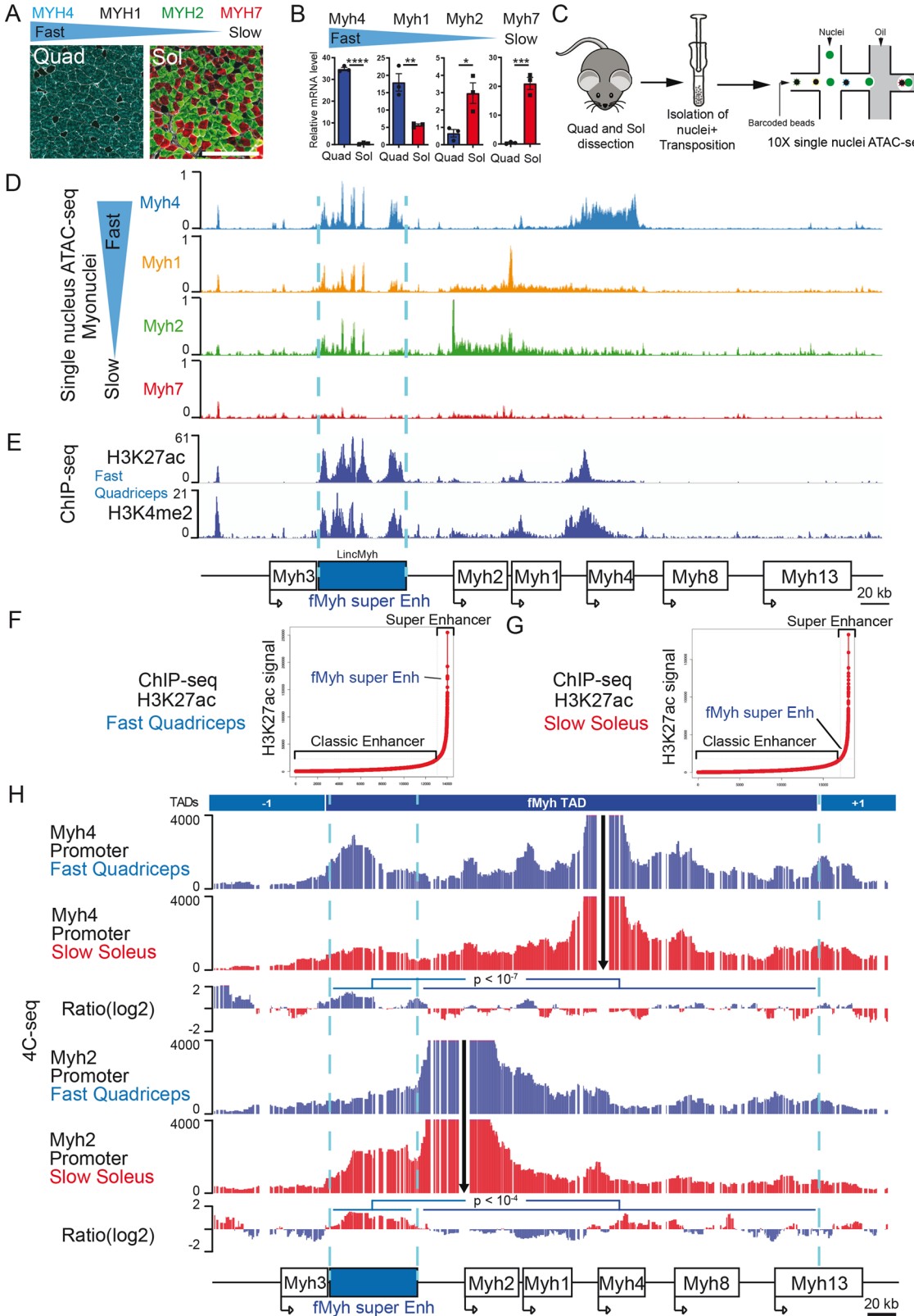

(Fig. S2B). These results show that the *fMyh*-SE establishes dynamic chromatin interactions with the promoters of the transcribed genes at the locus, with 3D spatial proximity directly coinciding with the activity of the promoters in each fiber type.

**The fMyh locus is organized in two topological associated domains**. In mammals, interactions between enhancers and promoters take place preferentially within topological associated domains (TADs) that are delimited by CTCF insulator binding sites. These CTCF sites can prevent enhancers from activating a gene present in another TAD[29–32]. TAD organization and CTCF insulator sites are conserved between cells and mammalian genomes[33]. We collected data of TAD organization in *fMyh* genes from available Hi-C experiments in embryonic stem cells[34]. As

**Fig. 1 Identification of a super enhancer in the intergenic region of _Myh3_ and _Myh2_. A** Adult myofibers express different MYH isoforms. Immunostaining against fast (MYH4, MYH2) and slow (MYH7) MYH of adult fast quadriceps (Quad) and slow soleus (Sol) muscle sections, MYH1 + myofibers by default appear black. **B** Quantification by RT-qPCR of _Myh_ mRNA expression in adult Quad and Sol ($n = 3$). **C** Graphical scheme of the experiments used for snATAC-seq experiments performed with slow soleus and fast quadriceps adult skeletal muscle. **D** Chromatin accessibility of the different types of myonuclei in the fast _Myh_ locus. In fast myonuclei, we identified a 42-kb region with multiple chromatin accessibility peaks in the intergenic region of _Myh3_ and _Myh2_ genes. In slow _Myh7_ myonuclei this region of chromatin is not accessible. **E** H3K27Ac and H3K4me2 ChIP-seq signals[25] were highly enriched in the 42 kb region of snATAC-seq peaks in the intergenic region of _Myh3_ and _Myh2_ genes. **F** Distribution of H3K27ac ChIP-seq signals across quadriceps and soleus enhancers[25]. SEs contain high amounts of H3K27ac and the f_Myh_ 42 kb sequence is identified as a SE. **G** Same as **F** in soleus. **H** 4C-seq experiments showing the interactions of the _Myh4_ (up) and _Myh2_ (down) promoters in quadriceps (blue) and soleus (red). Viewpoints are indicated by black arrows. The Ratio of interactions between the quadriceps and the soleus is indicated in between and shows that promoters of the active gene at the locus display significantly more interactions within the 42 kb cis-regulatory f_Myh_ super enhancer. Significance of difference: G-test. For **B**, significance of difference by Student _t_ test. Numerical data are presented as mean ± s.e.m. *$P < 0.05$, **$P < 0.01$, ***$P < 0.001$, ****$P < 0.0001$. Scale bars: 100 μm for **A**. Source data are provided as a Source Data file.

shown in Fig. S2B, the f_Myh_ genes are clustered in two distinct TADs separated by CTCF binding on boundary elements observed in ChIP-seq experiments[34]. One TAD includes the embryonically expressed _Myh3_, and another adjacent TAD includes all the other f_Myh_ genes. To confirm this 3D organization of the fast _Myh_ locus, we further analyzed 4C-seq experiments with different viewpoints all along the locus (Fig. S2B). These experiments confirmed that the _Myh3_ promoter interacted mostly with DNA sequences present in its TAD (−1TAD), while the other f_Myh_ promoters interacted almost exclusively with sequences present in the f_Myh_ TAD. We also observed that the f_Myh_-SE interacted mostly with sequences present in the f_Myh_ TAD (Figure S2B). This suggests that the adult f_Myh_ genes, the fetal _Myh8_ gene and the extraocular muscle-specific _Myh13_ gene, which are all located in the same TAD, could be controlled by the same f_Myh_-SE. On the contrary, either the regulatory element(s) that control the spatio-temporal expression of _Myh3_ should be distinct from the ones controlling the other f_Myh_ genes or the TAD boundary should be dynamically reorganized in cells where this gene is active.

**A transgenic mouse model of the f_Myh_ locus fully recapitulates _Myh1_, _Myh2_, and _Myh4_ expression.** To create a transgenic mouse model for f_Myh_ expression, we inserted the cDNAs coding for YFP at the ATG of _Myh2_, Tomato at the ATG of _Myh1_, and CFP at the ATG of _Myh4_ into a 222-kb bacterial artificial chromosome (BAC) that partially covered the f_Myh_ locus (end of _Myh3_ to the middle of _Myh8_) (Fig. 2A). A stop codon and a polyA tail were also inserted at the end of each transgene, preventing the expression of fusion proteins between cDNAs and the associated f_Myh_. The recombined BAC was injected in mouse oocytes and 2 separate transgenic animals were obtained and analyzed. We determined by qPCR on genomic DNA that one transgenic line called Enh+ integrated 2 complete copies of the entire length of the BAC including the SE. The second independent mouse line called Enh-, possesses an incomplete copy of the BAC devoid of the f_Myh_ SE (Fig. 2A). We observed efficient YFP, Tomato, and CFP expression in all skeletal muscles of Enh+ animals (Figs. 2B–D and S3A). Expression of the transgenes was not detected in the lung, liver, heart, or kidney (Figure S3B). Next, we compared the expression of the three transgenes with the accumulation of endogenous MYH proteins and mRNAs. As seen in Fig. 2E, YFP myofibers were detected in the slow soleus, in agreement with endogenous MYH2 expression, Tomato myofibers were detected in bracoradial muscles, and CFP myofibers in the quadriceps. By immunohistochemistry we observed a strong correlation between the expression of endogenous MYH2 proteins and YFP + myofibers, and between MYH1 proteins and Tomato myofibers (Fig. 2F, G). We did not observe the expression of the transgenes in slow MYH7 myofibers of the soleus

(Fig. S3C). This correlation between transgene and endogenous gene expression was confirmed by RT-qPCR. Efficient _YFP_ and _Myh2_ mRNA accumulation was found in the soleus of Enh+ mice. _Tomato_ mRNAs accumulated in both quadriceps and soleus like _Myh1_ mRNA. _CFP_ mRNA accumulated more in quadriceps than in soleus like _Myh4_ mRNAs (Fig. 2F–H). The three transgenes were detected in all skeletal muscles of the body including extraocular muscles and Esophagus (Fig. S4A, B). Tomato expression was first detected at P0 in the diaphragm when corresponding f_Myh_ genes expression become detected (Fig. S4C)[10]. Notably, most adult Enh+ myofibers expressed only one transgene although hybrid fibers[35] were also observed (Fig. S4D–F).

While efficient expression of CFP, Tomato, and YFP was detected in the skeletal muscles of Enh+ animals, very low expression of the three transgenes was observed in Enh- animals (Fig. 2I). This decreased transgene expression was also observed by immunostaining on adult muscle sections: much fewer YFP fibers in soleus and much fewer CFP fibers in gastrocnemius were detected in Enh- mice as compared to Enh+ mice (Figs. 2J and S5). Transgenes mRNA level was at least 100-fold lower in Enh- than in Enh+ animals, as estimated by RT-qPCR (Fig. 2K). Altogether, our results show that all regulatory sequences to fully recapitulate the spatiotemporal expression patterns of the f_Myh_ genes are present in the modified 222 kb BAC in Enh+ mice, which roughly overlaps the f_Myh_ TAD, and that the f_Myh_-SE and/or other sequences absent in Enh- transgenic animals are required for efficient _Myh2-YFP_, _Myh1-Tomato,_ and _Myh4-CFP_ transgenes expression.

Lastly, the Enh+ rainbow mouse line allows visualizing the fiber-type switches occurring during denervation, aging, in muscle-specific _Six1_ conditional knock out mouse models, and in other conditions at an individual scale (Fig. S6A–F) and is thus a powerful tool to study fiber-type changes in pathophysiological conditions[4,5].

**The f_Myh_-SE is required for adult f_Myh_ and neonatal _Myh8_ expression.** To assess the requirement of the SE for efficient f_Myh_ genes expression in vivo, we generated by CRISPR/Cas9 genome editing a knock-out mouse line deleted of this 42 kb element (Figs. 3A and S7A–B). Heterozygote mutant mice were viable and fertile and presented no obvious deleterious phenotype. In contrast, homozygote mutants died at birth, potentially due to impairment of respiratory skeletal muscle contractions as suggested by the absence of air in their lungs (Fig. 3B). E18.5 mutant fetuses showed no major visible skeletal muscle hypoplasia (Figs. 3C and S7C). In muscles of E18.5 mutant fetuses, the f_Myh_-SE deletion induced a strong decrease of the expression of adult f_Myh_ (_Myh2_, _Myh1_, and _Myh4_) and of neonatal _Myh8_ genes detected by RNAscope on isolated fibers from the diaphragm and

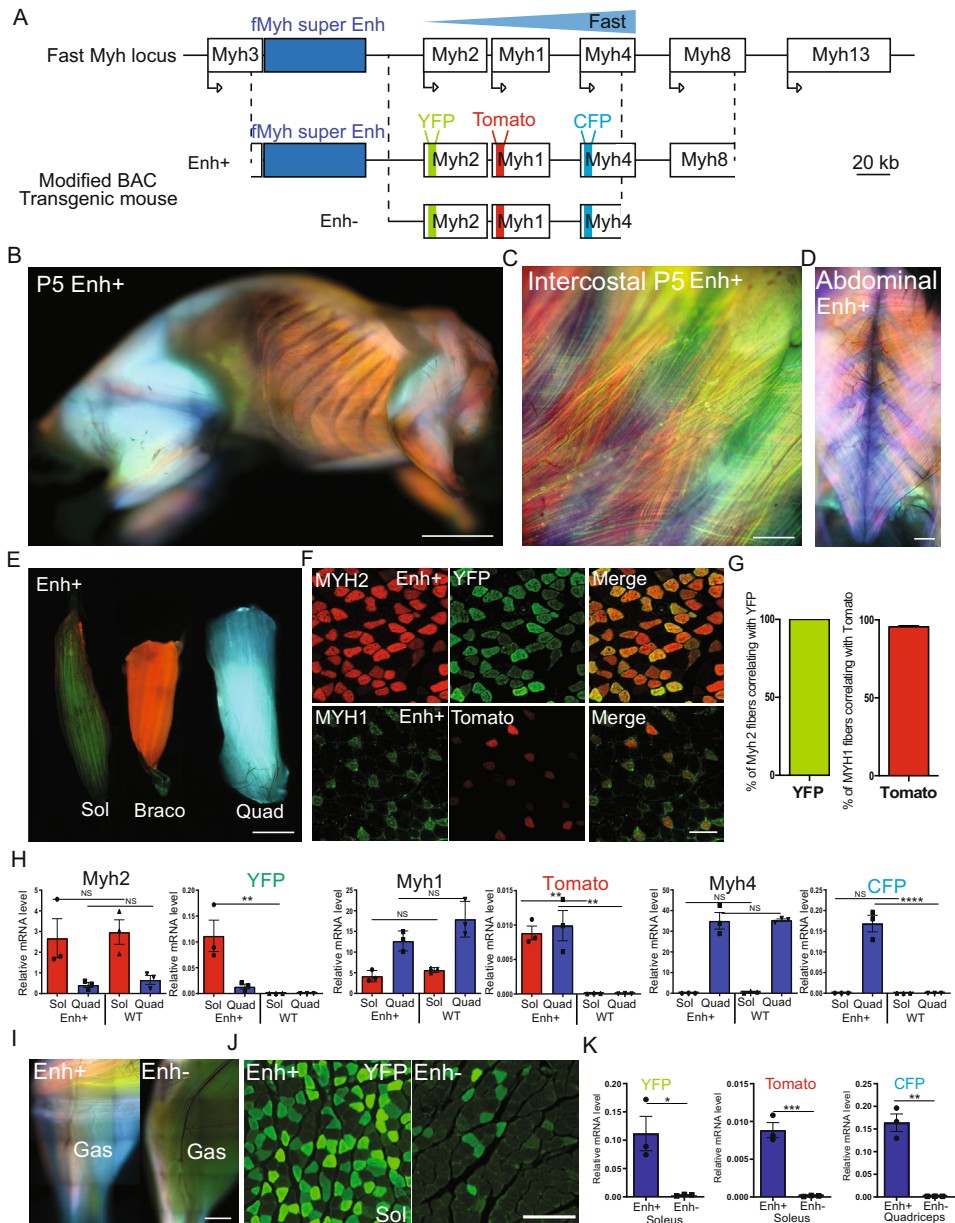

**Fig. 2 Transgenic models to study fMyh genes expression. A** Schematic representation of mouse f*Myh* locus and the recombined 222 kb Bacterial Artificial Chromosome (BAC) of the same locus. YFP, Tomato, and CFP cDNAs were inserted in the first exon of *Myh1*, *Myh2*, and *Myh4* genes respectively in the BAC. Two transgenic mouse lines were obtained, one called Enh + that integrated 2 complete copies of the BAC and the other called Enh- devoid of the SE region and the 3′ region of the locus. The transgenes YFP, Tomato, and CFP are not to scale. **B–D** Pictures of Enh + transgenic mice, red; Tomato, green; YFP and blue; CFP. All skeletal muscles expressed the transgenes. **B** 5-day-old lateral view. **C** zoom in intercostal muscles. **D** 2-month-old intercostal and abdominal muscles. **E** Transgene expression in adult soleus (Sol), bracoradial (Braco), and quadriceps (Quad) showing predominant expression of YFP in green, Tomato in red, and CFP in blue for each muscle. **F** Expression of the transgenes correlates with endogenous MYH protein expression in Enh+ line. Up: immunofluorescence against endogenous MYH2 (red) and of YFP (green) in adult soleus transverse section of Enh + mice. Down: immunofluorescence against endogenous MYH1 (green) and of Tomato (red) in adult quadriceps transverse section of Enh + mice. **G** Quantification of the percentage of MYH2 or MYH1 fibers expressing YFP or Tomato respectively, (*n* = 3). All MYH2 fibers are YFP + and almost all MYH1 fibers are Tomato+. **H** Relative expression level of mRNA in adult Sol and Quad of endogenous *Myh* genes and of transgenes, in wild type (WT) and in Enh + mice (*n* = 3). **I** Pictures of the adult leg of Enh + (left) and Enh- (right) mouse. The expression of the three transgenes is much higher in the Enh + line compared to Enh- mouse. **J** Immunostaining with GFP antibodies revealing YFP fibers on a section of adult Sol in Enh+ and Enh- mice. In Enh+ mouse, all MYH2 fibers expressed YFP whereas in Enh- only 10% of MYH2 fibers expressed YFP. **K** Quantification by RT-qPCR of transgenes expression in Enh+ and Enh− mouse line. Numerical data are presented as mean ± S.E.M. *P < 0.05, **P < 0.01, ***P < 0.001. Significance of difference, for **H**: two-way ANOVA and Student's *t* test for **K**. Scale bars: 100 μm for **F**, and 50 μm for **J**. Source data are provided as a Source Data file.

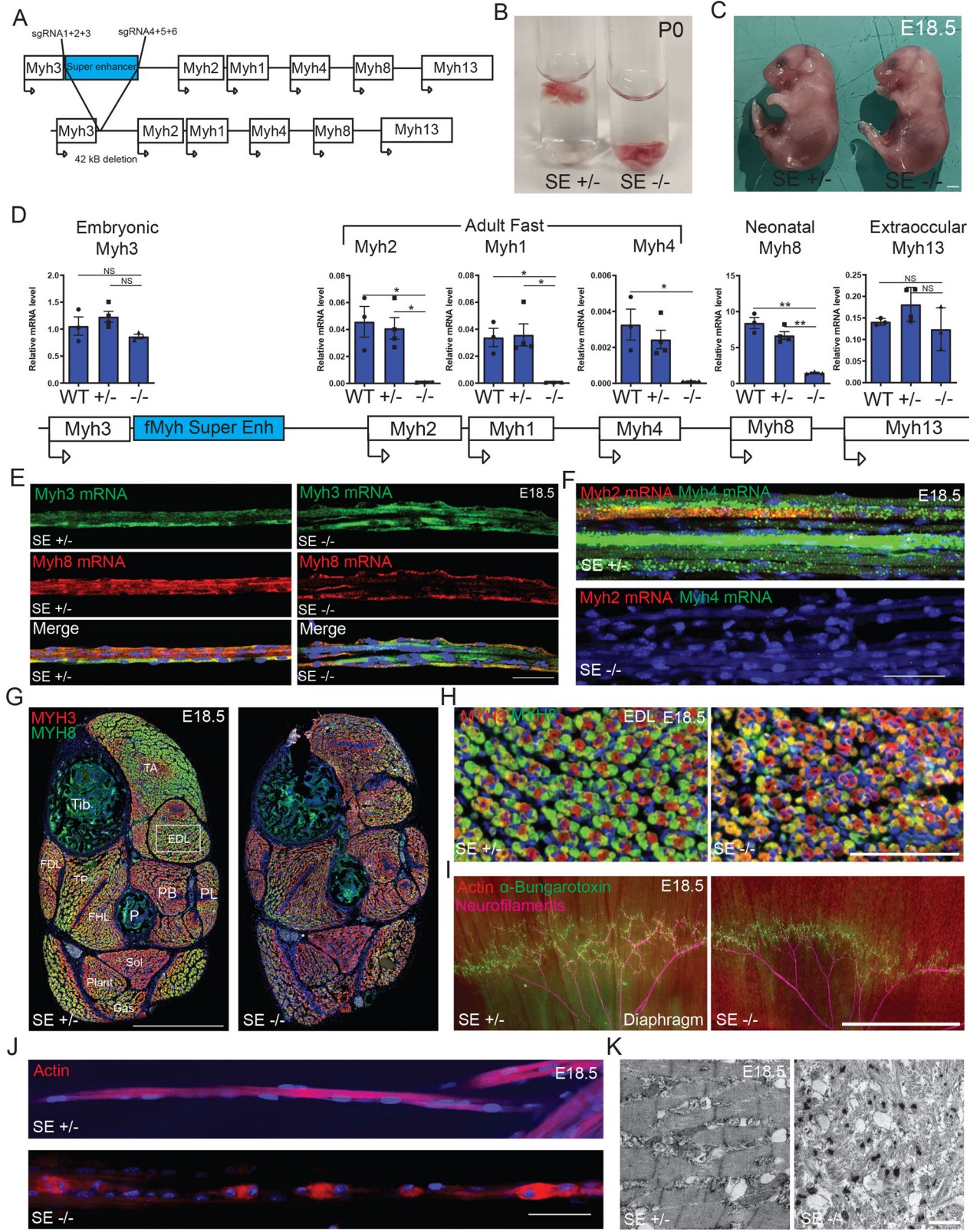

quantified by RT-qPCR on leg skeletal muscles (Fig. 3D–F). We detected at this embryonic stage regionalized low expression of adult f*Myh* along a few mutant fibers (Fig. S7D), indicating that the *Myh4* gene can be expressed in rare myonuclei in absence of the f*Myh*-SE. Thus, the f*Myh*-SE allows sustained expression of f*Myh* in the syncytium, although not all myonuclei at E18.5 have yet activated the expression of these adult forms[10]. *Myh4* or *Myh1*

simple KO[36,37] and *Myh1/Myh4* double KO (see below) mouse are viable and fertile. *Myh2* mutant mice have not been analyzed in detail but seem to have no major phenotype[38]. The absence of breathing and survival observed in P0 42 kb f*Myh*-SE mutants could be due to the loss of *Myh8* expression, or to the loss of the expression of a combination of several f*Myh* genes (Fig. 3D, G, H). At the limb level expression of *Myh3* and *Myh7* was not

**Fig. 3 The fMyh-SE is required for adult fMyh and neonatal Myh8 genes expression. A** A mouse line deleted for the fMyh-SE element was generated by injecting specific sgRNAs and Cas9 protein into mouse oocytes. **B** fMyh-SE$^{-/-}$ mice died at birth (P0) without breathing and air in their lungs. **C** fMyh-SE$^{-/-}$ E18.5 fetuses did not present severe visible malformations. **D** Quantification of Myh mRNAs by RT-qPCR in control and fMyh-SE$^{-/-}$ E18.5 forelimb skeletal muscles. Mutant muscles showed decreased Myh2, Myh1, Myh4, and Myh8 mRNAs levels. **E** RNAscope experiments against Myh3 and Myh8 mRNAs on isolated E18.5 forelimb fibers of control and mutant mice. **F** Same as **E** showing a decreased accumulation of Myh2 and Myh4 mRNAs in mutant mice compared to their littermate controls. **G**, **H** Immunostaining at the distal hindlimb level of E18.5 control and mutant fetuses revealing MYH3 and MYH8 positive myofibers. **H** zoom in the EDL of control and mutant fetuses. **I** In toto immunostaining of diaphragms from E18.5 mutant and control fetuses showing in red Actin filaments (phalloidine), in green AchR (alpha-bungarotoxin), and in pink neurofilaments. Mutant diaphragms show altered repartition of NMJ and punctated Actin aggregates. **J** Myofibers from mutant diaphragm showed defects in sarcomeres organization as shown by phalloidine staining. **K** Electronic microscopy pictures of the sarcomeres defects present in mutant E18.5 fetuses compared to their littermate controls. For **D** ($n = 3$). For **E** and **F**, scale bar: 50 μm. For **G**, scale bar: 500 μm. For **H**, scale bar: 100 μm, 500 μm for **I** and 25 μm for **K**. Numerical data are presented as mean ± s.e.m. *$P < 0.05$, **$P < 0.01$. Significance of difference, for **D**: one-way ANOVA with multiple comparisons. Source data are provided as a Source Data file.

affected as shown by RT-qPCR experiments and by immunocytochemistry against MYH3 and MYH7 (Figs. 3D, G, H and S7C). The mutant diaphragm myofibers were innervated, with however an abnormal distribution of neuromuscular junctions (Fig. 3I). In the 42-kb fMyh-SE E18.5 mutants many limb myofibers presented absence of sarcomeres associated with Actin aggregates around their myonuclei with only a few fibers that did not present these defects (Figs. 3J and S7E). We suspect that unaffected fibers could be primary fibers expressing Myh7 and or Myh3, whose expression appeared normal, while affected myofibers could be secondary myofibers that normally activate the expression of Myh8 (Fig. 3E). The absence of MYH8 could thus lead to sarcomere formation defects leading to Actin aggregates. Electronic microscopy experiments showed an accumulation of fibrillar materials in mutant diaphragm myofibers that may correspond to Actin accumulation in absence of MYH proteins (Fig. 3K), and the absence of sarcomere in many myofibers. No apparent tissue abnormalities were observed at the craniofacial level as revealed by MYH3, MYH8, Laminin and Dapi staining or at the axial level by HE staining (Fig. S7F, G). Altogether these results showed that the fMyh SE controls the expression of adult fMyh and neonatal Myh8 and that these isoforms are required for correct sarcomere formation in secondary myofibers and important for efficient muscle contraction at birth.

**The fMyh-SE is composed of distinct cis-regulatory modules (CRM).** SEs are composed of multiple enhancers and each with a specific role in promoter activation[39,40]. To characterize the role of two individual CRM defined by snATAC-seq experiments in the SE (Fig. S1B), we generated their deletion by CRISPR/Cas9 genome editing and obtained two distinct mouse mutant lines (Fig. S8A–C). The first CRM enhancer A (EnhA, matching to snATAC-seq peak 1, Fig. S1B) corresponds to a 5Kb region located at the most 3′ snATAC-seq peaks of the fMyh-SE (Fig. 4A). The second CRM enhancer B (EnhB, matching to snATAC-seq peaks 3, Fig. S1B) corresponds to two snATAC-seq peaks located in the middle of the fMyh-SE. We previously showed that this CRM can activate the expression of Myh1, Myh2, and Myh4 promoters in transient adult muscle transfection assays[23]. In contrast to homozygote mice deleted for the fMyh-SE that died at birth, we obtained viable and fertile adult EnhA and EnhB homozygote mutant mice. We determined the expression of MYH7, MYH2, and MYH4 in the distal hindlimb by immunohistochemistry (Fig. 4B) of these mutants. EnhA$^{-/-}$ mice showed a strong decrease of MYH4 expression in certain specific muscles (Fig. 4B, C). MYH4 was no more detected in the TP and the FHL limb muscles of EnhA$^{-/-}$, while the number of MYH1 fibers increased in these mutant muscles (Figs. 4B–D and S8D). This MYH4 fiber-type switch associated with the absence of the EnhA was also observed in other muscles (TA, EDL, PB, PL, FDL, and Plant) while other muscles (Gas and Sol) were spared. This result

was confirmed by RT-qPCR experiments showing downregulation of Myh4 expression in the TA of EnhA mutants (Figs. 4E and S8E). These results showed that enhancer A dominates regulation of Myh4 in specific muscles, probably through the recruitment of key Myh4 identity factors, while dispensable in others and showed also that MYH4 myofibers are not all equivalent. A low expression of Myh8 and Myh13 was also detected in adult WT TA which was strongly decreased in EnhA$^{-/-}$ TA, demonstrating that the expression of these two genes is also controlled by the enhancer A present in the SE (Fig. 4E).

In contrast to the EnhA mutant mice, we observed no major modification of slow MYH7 and fast MYH2 and MYH4 expression in muscles of EnhB mutant animals by immunostaining (Fig. 4B). As for EnhA mutant muscles, we observed a decrease of Myh8 and Myh13 expression in adult EnhB mutant TA compared to the WT (Fig. 4E). Linc-Myh expression was no more detected in EnhB$^{-/-}$ TA. We further generated a transgenic mouse line carrying an nls-LacZ transgene under the control of EnhB DNA sequences (Fig. S8F–G). Nls-LacZ transgene expression was detected only in fast and not in slow muscles. These results showed that even if the deletion of EnhB do not induce major alterations of adult fMyh expression, this DNA element has an enhancer activity in fast adult fibers, as already suggested[23]. Altogether analysis of these mutant mouse lines revealed that the SE is composed of distinct enhancer elements possessing distinct functions, two of which activate Myh1, Myh2, Myh4, Myh8, or Myh13 genes in specific muscles.

**The fMyh gene promoters compete for the SE.** To further elucidate the mechanisms controlling the specific and exclusive activation of fMyh promoters, we tested whether these promoters competed for the SE. A mouse model harboring a 72 kb deletion of the Myh1 and Myh4 genes (Myh(1–4)$^{Del}$) was generated by CRISPR/Cas9 genome editing (Figs. 5A and S9A–B). In the deleted allele, Myh8 and Myh13 genes are brought closer to the fMyh-SE, while the Myh2 promoter remains at the same distance from the fMyh-SE than in the WT allele. Myh(1–4)$^{Del/+}$ and Myh(1–4)$^{Del/Del}$ animals were viable. No expression of Myh1 and Myh4 was detected in Myh(1–4)$^{Del/Del}$ animals. These mutants presented a strong hypotrophy in specific areas of individual skeletal muscles, while other areas of the same muscle seemed preserved: the deeper regions of the TA and Gas were more spared than the superficial regions where small myofibers accumulated (Fig. 5B). This selective partitioning seemed to less affect deep muscles (Plantaris, PB) compared to the superficial areas of peripheral muscles like the TA or the Gas (Fig. 5B). In Myh(1–4)$^{Del/+}$ and Myh(1–4)$^{Del/Del}$ mouse, we observed increased Myh2 expression showing that the deleted allele for Myh1 and Myh4 does ectopically activate Myh2 in the deep regions of muscle masses (Figs. 5B, E, S9E, F, and S10). We also detected an increased expression of Myh8 and Myh13 in both Myh(1–4)$^{Del/+}$ and Myh(1–4)$^{Del/Del}$ mutant muscles (Figs. 5C–E

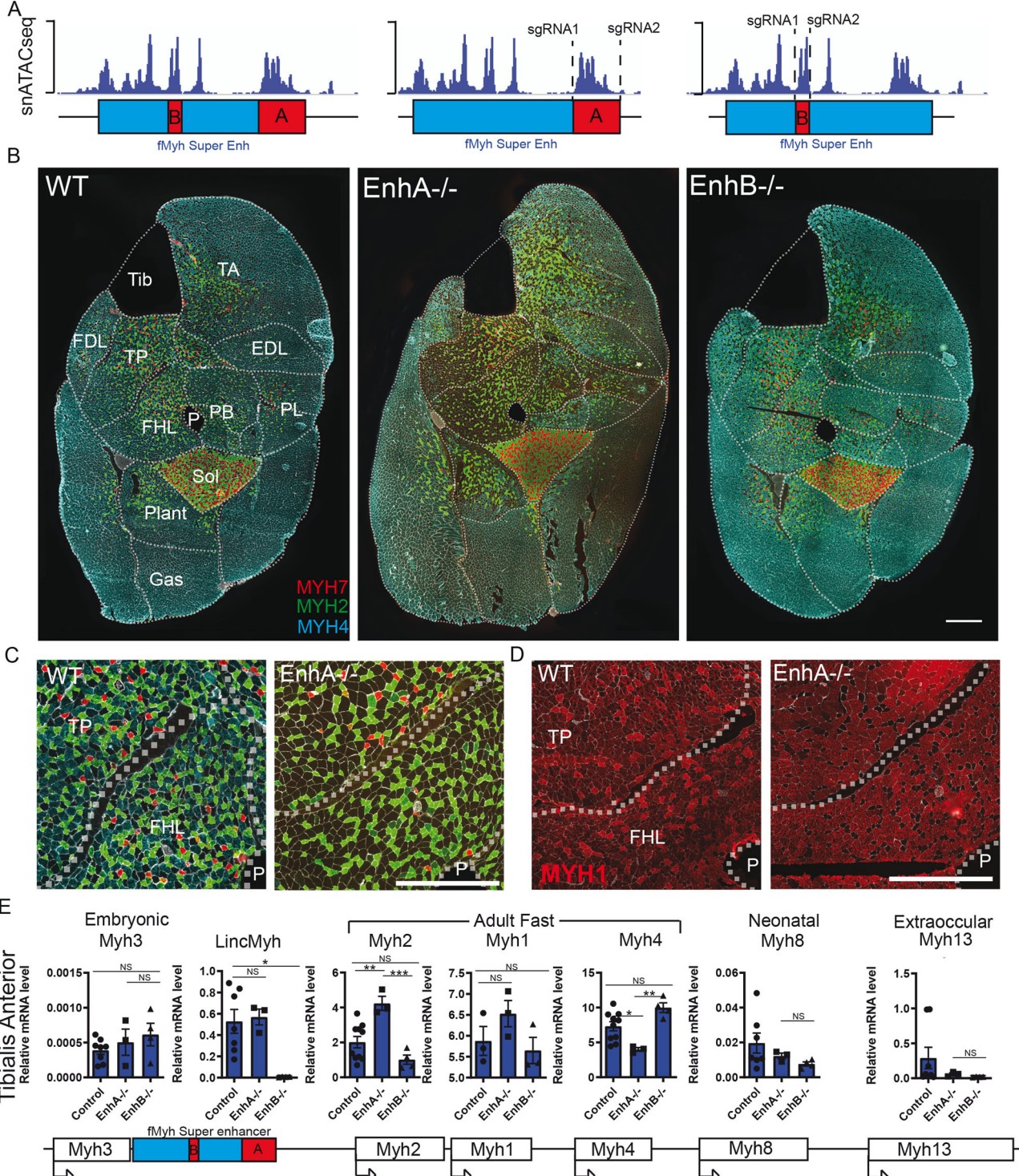

**Fig. 4 Role of the different enhancers composing the SE. A** Schematic representation of the snATAC-seq peaks along the 42 kb SE and the enhancers A and B deleted by CRISPR/Cas9 editing. **B** Immunostaining against fast MYH2, MYH4, and slow MYH7 on adult leg sections of 2–3-month-old mouse female deleted for enhancer A or B. **C** Same as **B**, zoom in Tibialis posterior and FHL muscle of WT and *EnhA⁻/⁻* mutant. **D** Immunostaining against fast MYH1 in Tibialis posterior and FHL muscle of WT and *EnhA⁻/⁻* mutant. The absence of *EnhA* induced an increased number of MYH1 positive fibers. **E** Quantification of f*Myh* mRNA and of *Linc-Myh* in adult TA of control and *EnhA* and *EnhB* mutant by RT-qPCR experiments. For **E**, n = 3. Numerical data are presented as mean ± S.E.M. *P < 0.05, **P < 0.01, ***P < 0.001. Scale bars: 100 μm for **B**–**D**. Significance of difference, for **E**: one-way ANOVA with multiple comparisons. Source data are provided as a Source Data file.

and S9F). Interestingly, we observed in the *Myh(1–4)^Del/Del* mutant a deep to peripheral gradient of MYH8 and MYH13 positive myofibers, with increased MYH13 fibers in the peripheral areas of muscle masses (Fig. 5C, D). Thus, in absence of *Myh1* and *Myh4* genes, the

f*Myh*-SE can activate the expression of either *Myh2*, *Myh8*, or *Myh13*, with a degree of plasticity of the myofibers depending on their position inside each individual muscle. These results show that each f*Myh* promoter competes for interaction with the SE and that

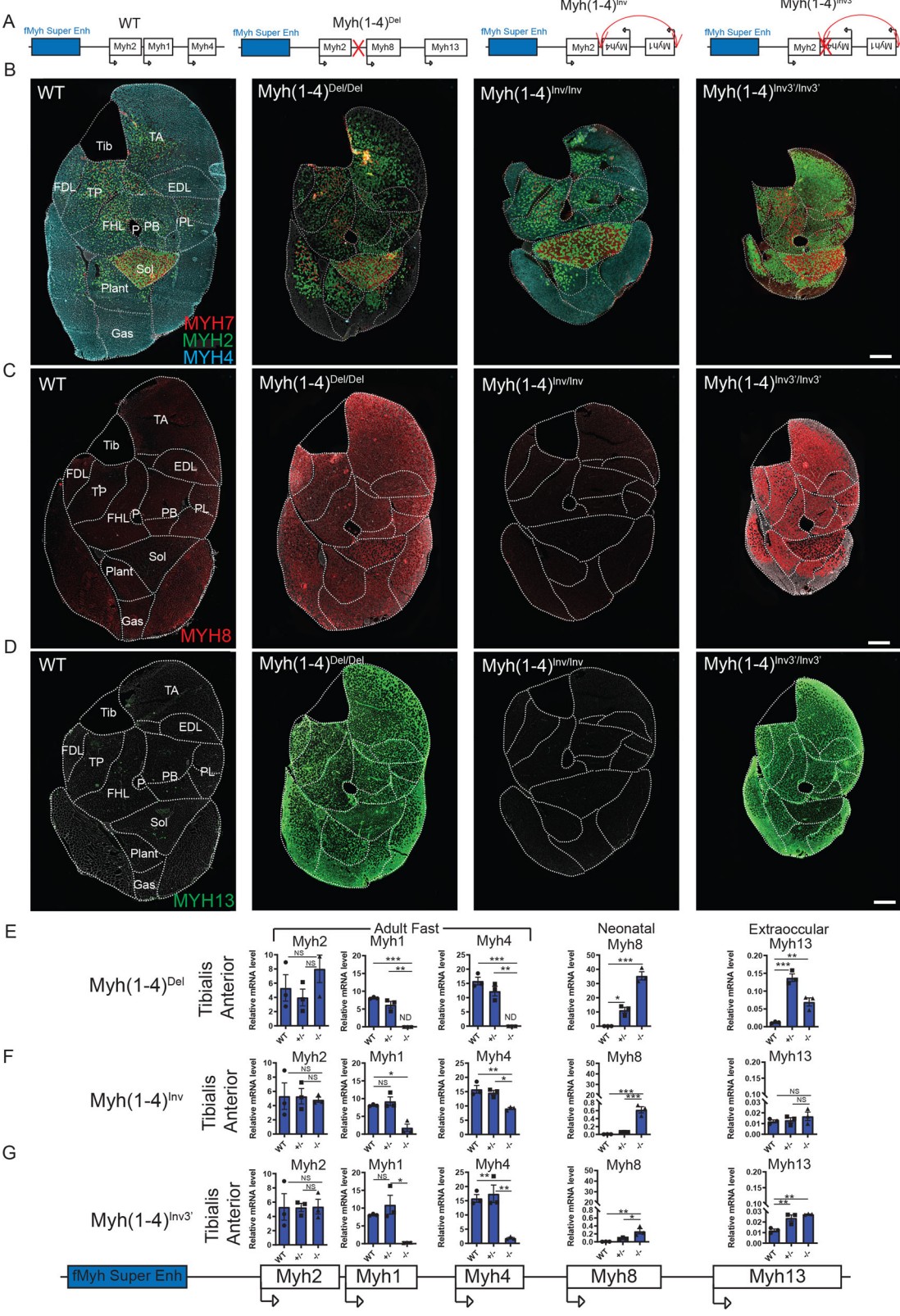

this competition is influenced by specific muscle sub volumes in agreement with a selective partitioning[41], and by the deep or superficial position of the muscle itself.

With the sgRNA used to delete *Myh1* and *Myh4*, we obtained two additional mouse lines with a complete inversion of *Myh1* and *Myh4* genes (*Myh(1–4)^Inv* and *Myh(1–4)^Inv3′*), allowing to test the hypothesis that the order of the *Myh1* and *Myh4* genes in

the locus is important for their correct expression. The distance between *Myh8* or *Myh13* and the SE was not modified in the *Myh(1–4)^Inv* allele compared to the WT allele. In both these mouse lines, the order of the f*Myh* genes in the locus was modified (*Myh2*, *Myh4*, *Myh1* then *Myh8*). The homozygote *Myh(1–4)^Inv/Inv* mutant mice were viable and showed a strong decrease of *Myh1* expression, a weaker decrease of *Myh4*

**Fig. 5 The promoters of f*Myh* genes compete for the shared SE. A** Schema of the distinct f*Myh* alleles generated by CRISPR/Cas9 editing. **B** Immunostaining against MYH4 (blue), MYH2 (green), and slow MYH7 (red) of adult distal hindlimb sections of WT, *Myh(1–4)$^{Del/Del}$*, *Myh(1–4)$^{Inv/Inv}$*, and of *Myh(1–4)$^{Inv3'Inv3'}$* mutants. **C** Immunostaining against neonatal MYH8 of adult leg sections in WT, *Myh(1–4)$^{Del/Del}$*, *Myh(1–4)$^{Inv/Inv}$*, and of *Myh(1–4)$^{Inv3'/Inv3'}$*. **D** Same as **C** against extraocular MYH13. **E** Quantification of *Myh2, Myh1, Myh4, Myh8,* and *Myh13* mRNAs of adult WT, *Myh(1–4)$^{Del/+}$* and *Myh(1–4)$^{Del/Del}$* TA by RT-qPCR experiments. **F** Quantification of *Myh2, Myh1, Myh4, Myh8,* and *Myh13* mRNAs of adult WT, *Myh(1–4)$^{Inv/+}$* and *Myh(1–4)$^{Inv/Inv}$* TA by RT-qPCR experiments. **G** Quantification of *Myh2, Myh1, Myh4, Myh8,* and *Myh13* mRNAs of adult WT, *Myh(1–4)$^{Inv3'/+}$* and *Myh(1–4)$^{Inv3'/Inv3'}$* TA by RT-qPCR. For **E–G** ($n = 3$). Numerical data are presented as mean ± S.E.M. *$P < 0.05$, **$P < 0.01$, ***$P < 0.001$. Significance of difference, for **C–E**: one-way ANOVA with multiple comparisons. Scale bars: 50 μm for **G**. Source data are provided as a Source Data file.

expression and no difference of *Myh2* expression compared to WT mice (Figs. 5B–D, F, S9C–E, G, and S10). This indicates that a closer proximity of the *Myh4* promoter to the SE did not increase its activity at the adult stage. The strong decrease of *Myh1* expression could be due to the increased distance between its promoter and the f*Myh*-SE, to the disorder of the genes at the locus, or more probably to missing elements in the *Myh1* promoter, since only 575 bp upstream of the transcription start site are associated with *Myh1* promoter in the inverted allele. We also observed an upregulation of *Myh8* in this mutant line (Fig. 5F).

In the other *Myh(1–4)$^{Inv3'}$* line, a deletion at the 3′ end of *Myh4* was observed, precluding MYH4 synthesis. The homozygote *Myh(1–4)$^{Inv3'/Inv3'}$* mutant mice were viable, but presented a severe skeletal muscle atrophy. In this mutant mouse line, we observed a strong decrease of *Myh1* and *Myh4* expression (Figs. 5B–D, G and S9E, H). Quantification of *Myh1* and *Myh4* pre-mRNA levels indicated that the transcription at the *Myh4* gene in TA was modestly decreased in *Myh(1–4)$^{Inv/Inv}$* and in *Myh(1–4)$^{Inv3'/Inv3'}$* mutant as compared with WT, while *Myh1* transcription level was severely downregulated (Fig. S9I, J). This showed that *Myh4* promoter can act as a decoy for the SE in *Myh(1–4)$^{Inv3'/Inv3'}$* since no MYH4 protein is produced. Similarly to the *Myh(1–4)$^{Del/Del}$* mouse line, we observed an upregulation of *Myh8* and *Myh13* expression in *Myh(1–4)$^{Inv3'/Inv3'}$* muscles (Fig. 5G). Interestingly in *Myh(1–4)$^{Inv3'/Inv3'}$* animals we observed many MYH2/MYH8 hybrid fibers and many pure MYH13 fibers preferentially in superficial areas of peripheral muscles like the TA or the Gas. MYH13 positive fibers were atrophic (Figs. 5D and 6C). We failed to detect MYH3 on *Myh(1–4)$^{Inv3'/Inv3'}$* adult hindlimb sections (not shown). These results showed that in the inverted allele, the SE could activate misoriented *Myh4* gene, but less efficiently, and activated the expression of *Myh8* and *Myh13* in the myofibers. Expression of MYH2 was detected all along the proximodistal axis in the distal hindlimb muscles of WT, *Myh(1–4)$^{Del/Del}$*, *Myh(1–4)$^{Inv/Inv}$*, and *Myh(1–4)$^{Inv3'/Inv3'}$* adult animals (Fig. S10), suggesting that myonuclei of mutant myofibers were similarly reprogrammed from one extremity of the muscle to the other. Altogether these results suggested that competition between the different *Myh* promoters for a shared SE controls their activation and that the order of the genes at the locus does not dictate their correct spatial expression.

**Limb skeletal muscles can be classified into three major categories with specific genetic programs.** Over 600 different skeletal muscles have been identified in the human body, each with a specific form, architecture, position, and function. In several myopathies, skeletal muscles can be specifically affected depending on their anatomic position[1]. Distinct genetic programs controlling the identity of each skeletal muscle in its specific environment may determine this selective vulnerability. The different mutants that were generated in this study presented distinct muscle phenotype depending on their location in the body. By comparing the fiber-type composition and fiber size in WT, *EnhA$^{−/−}$* and *Myh(1–4)$^{Inv3'/Inv3'}$* mutant mice (Fig. 6A–C),

we identified three different categories of skeletal muscles. The first category corresponded to muscles like the soleus, principally composed of small MYH7 and of MYH2 fibers (Fig. 6A). The soleus muscle was not affected in *EnhA$^{−/−}$* and *Myh(1–4)$^{Inv3'/Inv3'}$* mouse. The second category of muscles included muscles similar to the Tibialis posterior principally composed of MYH2, MYH1, and MYH4 fibers (Fig. 6B). These muscles were affected in *EnhA$^{−/−}$* and *Myh(1–4)$^{Inv3'/Inv3'}$* mutant mice and did not express MYH4 anymore. The last category of muscle regrouped muscles similar to the gastrocnemius expressing mainly MYH4 (Fig. 6C). The fibers of these groups of muscles presented a drastic decrease of fiber cross-section area in the *Myh(1–4)$^{Inv3'/Inv3'}$* mutants. In contrast, these muscles were not affected in *EnhA$^{−/−}$* mice. We next extended this study in proximal and distal muscles of the fore- and hindlimbs (Fig. 6D–G). As observed at the distal hindlimb level, muscles in forelimbs and proximal hindlimb showed distinct phenotype depending on their deep or superficial position[42]. We could detect specific localization of these three groups of muscles in the different parts of the hindlimb and forelimbs but with spatial patterns that seemed similar. The category of muscles with similar properties to the Soleus (shown in red) was the most internal in the limb. In contrast, the category of muscles with similar properties to the Gastrocnemius (shown in blue) was the most external. The category of muscles similar to the Tibialis posterior (shown in green) was located between these two groups (Fig. 6D–G). In the proximal part of the hindlimb, the group of muscles shown in blue was the most important and were severely affected in *Myh(1–4)$^{Inv3'/Inv3'}$* mutants, whereas the same group of muscles was almost not affected in *EnhA$^{−/−}$* mutants (Fig. 6E). In the distal part of the forelimb, the group of muscles shown in green prevailed over the other (Fig. 6F) whereas in the proximal part, the distribution of these muscles groups was more heterogeneous (Fig. 6G). Altogether these results revealed that limb skeletal muscles could be classified into 3 major categories with distinct properties and possessing different codes of transcription factors controlling their plasticity.

## Discussion

In adult muscles the contraction and general metabolic properties of the specialized myofibers are dictated by the expression of specific slow MYH7 and fMYH subtypes (MYH2, MYH1, MYH4, MYH13)[4,5]. Transient transfection experiments of GFP reporters previously suggested that the proximal (800–1000 bp) promoters of the *Myh2, Myh1,* and *Myh4* genes were sufficient to drive their spatial expression in adult muscles[43]. By combining single-nucleus ATAC-seq, ChIP-seq, and 4C-seq data from adult fast and slow skeletal muscles, we show here that f*Myh* genes, with the exception of *Myh3*, are regulated by a shared super enhancer. In fast-type myonuclei this SE interacts dynamically with the activated promoters of the locus by 3D chromatin looping. By using rainbow transgenic mouse models of the locus and knock-out mouse models of the SE, we show that this SE controls the level and the spatio-temporal specificity of f*Myh* genes expression in myonuclei and myofibers through exclusive interactions with their promoters. By disrupting the organization of the f*Myh* locus, we uncover positional heterogeneity within limb skeletal muscles

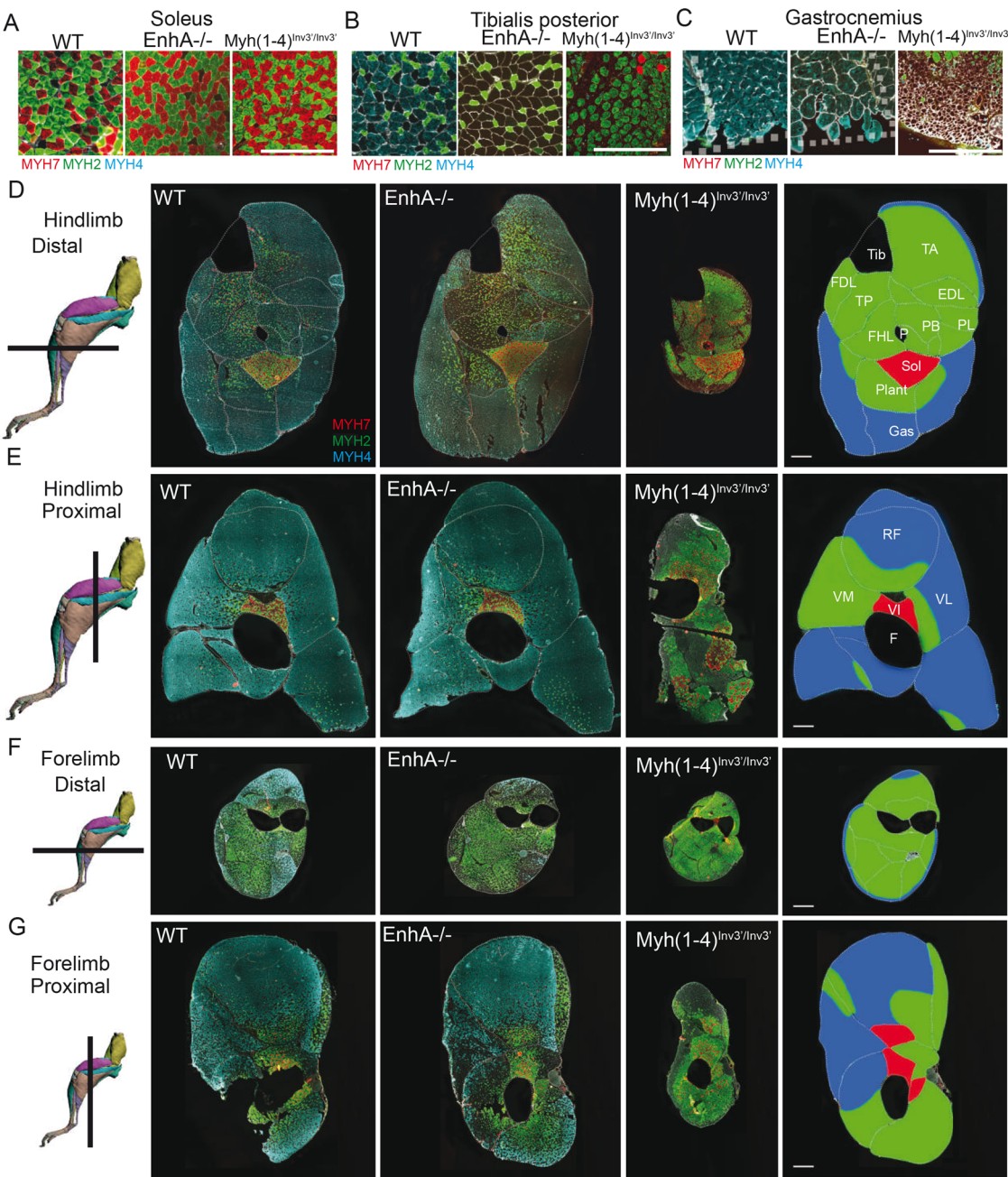

**Fig. 6 MYH expression established different groups of limb skeletal muscles in mutant animals. A–C** Immunostaining against MYH7 (red), MYH2 (green), and MYH4 (blue) in WT, *EnhA*$^{-/-}$, and *Myh(1-4)*$^{Inv3'/Inv3'}$ adult mice. The soleus (**A**) is not affected in these mutant mice. In the Tibialis posterior (**B**), the expression of MYH4 is lost in *EnhA*$^{-/-}$, and in *Myh(1-4)*$^{inv3'/inv3'}$ whereas an upregulation of MYH1 is observed in *EnhA*$^{-/-}$ mice and an upregulation of MYH2 in *Myh(1-4)*$^{Inv3'/Inv3'}$ mice. In Gastrocnemius (**C**), the peripheral fibers of *Myh(1-4)*$^{Inv3'/Inv3'}$ mice present a severe atrophy. In contrast these fibers are not affected in *EnhA*$^{-/-}$ mouse. **D** Comparison of the phenotypes of adult muscles in *EnhA*$^{-/-}$ and in *Myh(1-4)*$^{Inv3'/Inv3'}$ allowed classification of distal hindlimb muscles in three major categories. The first group is shown in red corresponded to the soleus that is not affected. The second group shown in green corresponded to muscles affected in *EnhA*$^{-/-}$ and *Myh(1-4)*$^{Inv3'/Inv3'}$ mutants. The third group shown in blue corresponded to muscles strongly affected in *Myh(1-4)*$^{Inv3'/Inv3'}$ but not in *EnhA*$^{-/-}$ mutants. **E** Same as **D** in the proximal part of the hindlimb. **F** Same as **D** in the distal part of the forelimb. **G** Same as **D** in the proximal part of the forelimb. For **D**, scale bar: 100 μm. **D–G**: drawings of hindlimbs and forelimbs are from Charles et al.[42].

that may underlie selective muscle vulnerability observed in certain human neuromuscular diseases.

We showed that a BAC containing 250 kb of DNA of the f*Myh* locus, from the 3′ end of the *Myh3* gene to the middle of the *Myh8* gene, recapitulates the endogenous spatiotemporal expression of *Myh2, Myh1,* and *Myh4* genes, while a shorter BAC devoid of the SE does not. Expression of the *Myh1*-Tomato

transgene was detected from P0 in the diaphragm, and expression of all three transgenes from P5 in most skeletal muscles, a period during which adult *Myh* endogenous genes are upregulated and relay *Myh8* expression[8,44]. This change in expression at the f*Myh* locus is recapitulated in the BAC transgenic mice where expression of the *Myh2*-YFP, *Myh1*-Tomato, and *Myh4*-CFP transgenes is detected from postnatal to adult stages. Most adult myofibers

express a single transgene, but hybrid fibers were also detected, mainly in soleus muscles, in agreement with previous findings in adult mouse muscles[10,35]. Furthermore, we showed that the expression of the three fluorescent reporter proteins provides a good readout of the fiber type modifications occurring during ageing, in male and females, after nerve crush or in *Six1* mutant animals[4,5]. All DNA sequences required for efficient *Myh2, Myh1,* and *Myh4* expression and for their mutual interactions are thus present in this 250 kb region. This muscle-rainbow transgenic mouse will therefore be a useful tool to image in vivo the contraction properties of specific fast myofiber subtypes in pathophysiological conditions when fiber type modifications occur[45–47].

SEs, which are composed of multiple enhancers, allow a more efficient recruitment of coactivators than conventional enhancers. During this process, multimolecular assemblies form by liquid-liquid phase separation, allowing aggregation of the transcriptional machinery in membraneless nuclear droplets[16,24]. Known SEs have been described to achieve a relatively constant high transcriptional activity, contrasting with the transcriptional bursts provided by typical enhancers that lead to episodic gene expression[15,48].

Here we show that the fine spatio-temporal expression of the f*Myh* genes is governed by a SE, which interacts with f*Myh* promoters by 3D chromatin looping, and is engaged in exclusive interactions with a single *Myh* promoter at the locus. Previous data of RNAscope experiments with f*Myh* premRNA probes demonstrated the coordinated firing of both alleles of specific f*Myh* genes in adult myonuclei[10]. These finding imply that selective SE-promoter loops may form simultaneously on both alleles of a given f*Myh* gene in the majority of body nuclei of each adult myofiber allowing sustained bi-allelic expression of a single gene, while the expression of the other genes at the locus is undetectable. Altogether, these results show that the f*Myh*-SE activates a single f*Myh* gene at the f*Myh* locus, suggesting the f*Myh*-SE cannot simultaneously activate two f*Myh* promoters and arguing against the existence of flip-flop mechanisms between the f*Myh*-SE and the different promoters of the locus as proposed earlier[17,49,50]. Whether this apparently non-stochastic gene expression in adult myofibers is true for all muscle genes governed by a SE remains to be established. At the f*Myh* locus, we suspect that these exclusive interactions between the SE and specific promoters are responsible for the high level of f*Myh* expression and to prevent the expression of two different MYH in adult myofibers (Fig. 7). To test if these exclusive interactions result from a competition between the SE and the associated f*Myh* promoters, we analyzed the consequences of *Myh1* and *Myh4* deletion. Muscles of adult *Myh(1–4)^(Del/Del)* mutant were composed of myofibers expressing MYH2, MYH8, or MYH13. Remarkably, *Myh2*, which is closest to the SE, was upregulated only in the deep regions of skeletal muscles, while in more peripheral myofibers where *Myh4* is normally predominantly expressed *Myh8* or *Myh13* were activated. These results show that *Myh2* cannot be activated in these peripheral myofibers, even in absence of *Myh1* and *Myh4,* and that competition between the promoters varies depending on the muscle position inside the limb, probably due to the differential enrichment of specific transcription factors in deep and peripheral muscles. These experiments demonstrated that some myofibers have the ability to switch from one specific promoter to another non-random promoter, suggesting that the transcription factors bound to *Myh8* and *Myh13* promoters in adult WT limb myofibers are able to interact with the SE, but compete less efficiently than those bound to *Myh4*. This is probably due to a lower frequency of interactions. In adult WT limb muscles these preferential interactions concur to favor *Myh4* at the expense of *Myh8* and *Myh13*

expression. In addition, even in *Myh(1–4)^(Del/Del)* mutant, very few hybrid fibers were detected[10,35], suggesting that most nuclei within each fiber activated a single gene, and that the SE was still contacting a single f*Myh* promoter in an exclusive manner. Such exclusive interactions were also detected at the ß-*Globin* locus where the LCR/SE interacts with a single promoter and where the order and the distance between the LCR and the *Globin* genes dictates their temporal expression[20,51].

We cannot formally exclude from our experiments that all f*Myh* genes, including the inactive promoters at the locus, are associated in a phase-separated droplet where all promoters interact with the SE in a common nuclear compartment, like at the α-*Globin* locus[52]. In this hub model associating all f*Myh* genes at the locus, the "inactive" promoters would be bound by a low amount of TFs that were not detected in snATAC-seq experiments, and associated with a low transcript level undetected in RNAscope experiments or by YFP, CFP, Tomato expression in the transgenic BAC model. In our preferred competition/exclusion model, inactive promoters at the locus are not associated in the phase-separated droplet due to the low amount of TF bound to their DNA sequences, while a specific promoter bound by multiple TFs and cofactors in certain myonuclei can be committed to the condensate (Fig. 7). In this model, deletion of the *Myh1* and *Myh4* genes in myonuclei destined to express these genes would allow the continuous expression of *Myh8* from late fetal to adult stage, due to the loss of competition with the *Myh1* and *Myh4* promoters for the SE. This competition could occur during postnatal development in WT animals to switch from *Myh8* expression to the expression of a single adult f*Myh* gene in a given myonucleus[10]. The transcription factors and cofactors involved in the switch from *Myh8* to *Myh2, Myh1* or *Myh4* remain to be characterized. We showed earlier that SIX1 homeoproteins are required for *Myh2* expression in the soleus, their absence precluding f*Myh* gene expression in this muscle, while reducing *Myh4* expression in other muscles[53,54]. The abundance of MEF3 sites binding SIX transcription factors in the 42KB SE[23] suggests that this protein family is a good candidate with their associated EYA cofactors to participate in a phase-separated droplet at the f*Myh* locus to drive efficient gene expression of f*Myh* genes at the locus[10].

Another element that may contribute to the fine-tuning of the expression of f*Myh* genes within the locus is the distance separating their promoters and the SE. During development, *Myh8*, remotely located from the SE, is preferentially activated compared to the other genes at the locus, arguing that the SE does not systematically interact with the closest promoter. Reciprocally, *Myh2*, which is closest to the SE, is expressed in far fewer adult myofibers than *Myh4*, which is activated in a majority of hindlimb muscles. However, the importance of gene location in relation to the SE might be modulated by the general context at the locus. Deletion of the *Myh1* and *Myh4* genes (*Myh(1–4)^(Del/Del)*) brings *Myh8* and *Myh13* closer to the SE. This proximity could participate in the upregulation of these two genes in *Myh(1–4)^(Del/Del)* animals, although competition could also account for the observed effects. In the inversion models, which conserve the distance between the SE and *Myh8* and *Myh13* as in the wt allele, *Myh8* and *Myh13* upregulation were more modest than in the deletion model. Still, in the deletion model *Myh8* and *Myh13* activation occurred in far more muscle fibers than in fibers where the *Myh2* gene is activated, again indicating that proximity with the SE is not the sole rule governing SE-promoter interactions at the locus. A mouse model with a permutation of the *Myh4* and *Myh8* genes could definitively address the importance of gene distance from the SE for their activity.

To precise the role of the potential enhancer elements composing the SE, we focused on two elements, enhancer A and

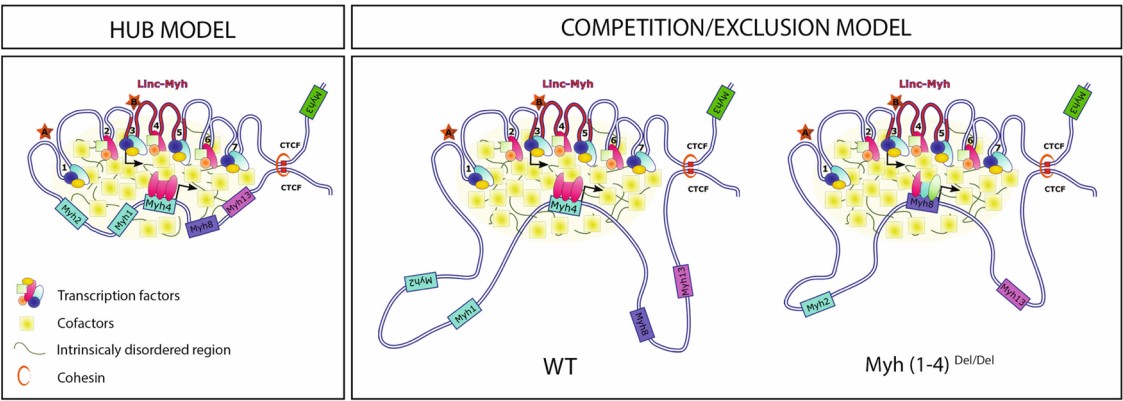

**Fig. 7 Two models explaining the complex regulation of *fMyh* genes by the shared SE.** The SE is composed of seven enhancer elements (1–7) recruiting TF and cofactors allowing the nuclear formation of a phase separation condensate in myonuclei and allowing robust *fMyh* expression, adapted from Sabari et al.[16]. Left, in the hub model even the inactive promoters at the locus are associated in the phase-separated droplet in *Myh4* + myonuclei. Right, in the competition/exclusion model, *Myh2*, *Myh1*, *Myh8,* and *Myh13* are excluded from the phase-separated droplet in *Myh4* + myonuclei because they are not bound by sufficient amount of TF and cofactors. In those myonuclei robust bi-allelic expression of *Myh4* is achieved, while transcription of the other genes is not detected[10]. In this competition/exclusion model, the deletion of *Myh1* and *Myh4* could lead to the maintenance of *Myh8* expression through development in specific muscles, (while in others *Myh13* is activated (not represented)) due to the absence of competition by *Myh4* or *Myh1* promoters. *Linc-Myh* spans enhancers 3–5. The models are not to scale.

enhancer B. Deletion of enhancer A led to an upregulation of *Myh1* in myofibers of peroneal muscles, without upregulation of the nearest *Myh2* promoter, again suggesting that the SE deleted of enhancer A still contacts a single fMyh promoter at a time with physically exclusive interactions. In the TA we observed an upregulation of *Myh2* in the enhancer A mutant with a decrease in *Myh4* expression, while *Myh2* was not upregulated in the Soleus. These results suggest that element A is not bound by negative TF, but that element A is bound by positive TF in *Myh4* + myonuclei of specific muscles to associate *Myh4* within the SE, allowing its high expression level at the expense of the other *Myh* genes at the locus. The A-mutant SE-*Myh4* interactions are weaker than those observed with the WT SE, while the interactions between the A-mutant SE and *Myh2* become predominant, ensuring the transcription of this gene at the expense of *Myh4* in specific myofibers. Down-regulation of *Myh4* in enhancer A mutant was observed only in certain muscles. This implies that the SE is composed of individual enhancer elements that may have redundant activities under the control of muscle identity genes, as in *Drosophila*[55,56]. This redundancy may contribute to the expression of a single gene at the locus.

Deletion of enhancer B in the SE led to the complete absence of *Linc-Myh* gene expression demonstrating that this long non-coding nuclear RNA is not required in distal hindlimb muscles to achieve efficient *Myh4* or *Myh1* expression, in contrast to what observed after its knock down by shRNA in adult TA[23]. Nevertheless, the expression level of *Myh2*, *Myh8,* and *Myh13* was decreased in TA and soleus of enhancer B mutant animals showing its activity in adult myonuclei. Enhancer B itself was able to drive *nls-LacZ* transgene expression in adult myonuclei of fast muscle as shown by additive transgenesis supporting its role as a muscle-specific enhancer element. Interestingly, deletion of the entire *Linc-Myh* gene corresponding to a 8 kb region deleting two snATAC-seq peaks (elements 4 and 5, 5′ to element B, Fig. 7) had as well no major impact on MYH1, MYH2, or MYH4 positive myofiber number in the TA or soleus[57], suggesting a modest role of this long non-coding RNA and of enhancers 4 and 5 in the SE activity in adult limb myofibers. Altogether, we did not identify within the SE a specific enhancer responsible for driving the expression of either *Myh1*, *Myh2*, or *Myh4* in all skeletal muscles. We cannot exclude that analysis of deletion of other snATAC-seq peaks may reveal a "master" enhancer element inside the SE.

Alternatively the *fMyh*-SE may be composed of redundant elements, none of which absolutely required for driving efficient and specific gene expression, as fund at the *α-globin* locus and for enhancers controlling limb and digits morphogenesis where partially redundant enhancers are suspected to provide both flexibility and robustness of gene expression[39,58–61]. A similar mode of regulation has been shown for the *Myf5/Mrf4* locus, where multiple enhancer elements are involved in the regulation of these two genes in the head, neck, back, thoracic and limb muscles during embryonic development and in the adult[62]. This illustrates the complexity of skeletal muscles gene regulation driving their diversity according to their localization and function in the body.

Our experiments reveal the importance of the *fMyh*-SE for muscle integrity and function. Deletion of the SE induced impaired ability to breathe leading to death at birth. This deletion impaired *Myh1*, *Myh2*, *Myh4*, and *Myh8* gene expression in skeletal muscles of E18.5 fetuses (one day before birth in C57BL/6N mouse strain), demonstrating the involvement of the SE to control their expression. Absence of *Myh8* expression may be involved in the death of the mutant animals. The requirement of *Myh8* expression during fetal development for efficient muscle contraction at birth is supported by the phenotype of *Myod*[−/−];*Nfatc2*[−/−] mice, where *Myh8* is no more expressed in intercostal muscles. These mutant mice do not survive after birth due to their inability to breathe[63]. In agreement we identified strong sarcomerisation defects associated with Actin aggregates in *fMyh*-SE[−/−] E18.5 mutant myofibers at the limb and diaphragm level, suggesting a complete absence of MYH and their inability to contract. The SE is present as well in the human *fMYH* locus and could be involved in the control of the *fMYH* genes as in mice. *MYH8* and *MYH2* are expressed during human fetal development, *MYH1* is detected after birth[8,64], while *MYH4* is only expressed in extra ocular myofibers due to mutations in its promoter region[37,65]. Absence of *MYH2* is associated with early onset myopathy characterized by mild generalized muscle weakness with predominant involvement of muscles of the lower limbs, and by ophtalmoplegia[66]. In contrast, *MYH1* mutations have not yet been reported and *MYH8* mutations do not seem to be associated with trismus-pseudocamptodactyly[67], contrarily to what was previously suspected. Mutations or deletions of the *fMYH*-SE have not yet been identified in human pathologies. Congenital myopathies can be associated with Actin aggregates,

fiber type disproportion or arthrogryposis[68–70], but not all these myopathies have been characterized at the genetic level.

This positional heterogeneity of skeletal muscles is reflected in certain neuromuscular diseases by a spectrum of clinical manifestations, with some muscles affected while other are spared, depending on the pathology[1]. As mentioned above, distinct signaling pathways and TFs are involved in the acquisition of the myogenic fate of progenitors depending on their anatomical position, which may underlie the susceptibility of specific muscles or groups of myofibers to environmental or genetic alterations[45,71,72]. Whole-body magnetic resonance imaging and muscle ultrasound in patients affected by Collagen VI deficiency, Dystrophin deficiency, or in ALS showed that specific muscles or specific group of myofibers inside a muscle mass can be specifically affected, while others are spared[45,73–75]. Little is known about the mechanisms driving this variability in susceptibility and understanding the underpinning mechanisms is a major challenge to develop adapted targeted therapies. By disrupting the organization of fMyh at the locus, we uncovered positional heterogeneity within limb skeletal muscles and defined three major categories of limb muscles. These three categories of stereotyped muscles are differentially positioned in the distal and proximal forelimbs and hindlimbs and illustrate that all $Myh4+$ myofibers are not equivalent. Such diversity depending on position has been observed clinically in Collagen VI deficiency, Dystrophin deficiency or in ALS. Based on snRNA-seq experiments on adult mouse muscles, we suspect that positional heterogeneity may be the consequence of distinct genetic programs that lead to the activation of groups of genes associated with either $Myh4$, or $Myh1$ or $Myh2$ expression. Indeed snRNA-seq analysis[10,76–78] has revealed an unsuspected genetic variability in $Myh4+$ and other myofiber types, with at least three subclasses of $Myh4+$ myonuclei and several subclasses of $Myh1+$ and $Myh2+$ myonuclei in mouse hindlimbs. Whether this diversity is at the origin of the deep/superficial gradient of muscle susceptibility observed in the present study and in certain neuromuscular diseases remains to be precisely tested.

## Methods

**Animals**. Animal experimentations were carried out in strict accordance with the European STE 123 and the French national charter on the Ethics of Animal Experimentation. Protocols were approved by the Ethical Committee of Animal Experiments of the Institut Cochin, CNRS UMR 8104, INSERM U1016 and by the Ministère de l'éducation nationale de l'enseignement et de la recherche, APA-FIS#15699-2018021516569195. We used 6–8 weeks old C57BL/6 N mouse female for most of our experiments. 6–8-weeks-old C57bl6N females were used in this study. Mice were maintained at temperature 22+/−2 °C, with 30 to 70% humidity and with a dark/light cycle of 12 h/12 h. Mice were anesthetized with intraperitoneal injections of ketamine and xylazine and with subcutaneous buprecare injections before denervation which was performed by sectioning of the sciatic nerve in one leg. All efforts were made to minimize animal suffering, and to reduce the number of animals required for the experiments.

**BAC targeting constructs and Myh locus modifications**. For the construction of the targeting vector pGEM-T-EasyMyh2YFP, C57BL/6N mouse DNA was first used as a template to clone 5′ arm and 3′arm of Myh2 with forward 5′- GAA TGA TTT CAT TGC TAC TTC -3′ and reverse HindIII 5′- GCT CAT GAC TGC TGA ACT CAC -3′, and forward HindIII 5′- AGT CCG AAA AGG AGC GAA TC -3′ and reverse 5′- GGT GAC TTC TAG TGA CTG AG -3′, respectively. The 5′ arm and 3′arm fragments were then cloned into a pGEM-T-Easy vector with HindIII in-between to make pGEM-T-EasyMyh2. The Yellow Fluorescent Protein (YFP) coding sequence was PCR amplified (PHUSION, Thermofisher) and cloned in pBluescriptSK+ using EagI-XbaI sites provided by the primers. Fragments containing three polyA sequences (rabbit β-globin, HSV-TK, and BGH) and LoxP-kanamycin-LoxP were then extracted from preexisting constructs and introduced downstream of YFP. The whole YFP-3pA-LoxP-kana-LoxP fragment was amplified (PHUSION, Thermofisher) with forward 5′- CAG CAG TCA TGA GCA TGG TGA GCA AGG GCG AGG AG-3′ and reverse 5′- CTC CTT TTC GGA CTA CGA CTC ACT ATA GGG CGA ATT G-3′ primers. The resulting amplicon features 15 bp homology in 5′ and 3′ extremities with the targeting arms allowing Sequence and Ligation Independant Cloning (SLIC) into the HindIII digested

pGEM-T-EasyMyh2 plasmid (GeneArt Seamless Cloning and Assembling kit, Thermofisher).

Similarly, for the construction of the targeting vector pGEM-T-EasyMyh1Tomato, targeting arms were PCR generated from C57BL/6N mouse DNA and assembled together with HindIII in-between (pGEM-T-Easy-Myh1: 5′ arm forward 5′- CAT CCA GCA TGT GTT CTC AGA GGT -3′, reverse HindIII 5′- ACT CAT GGC TGC GGG CTA TT -3′; 3′arm forward HindIII 5′- GTC TGA AAA GGA GCG AAT CGA G -3′, reverse 5′- AGT AGG TCT GCA TCA AGA GAG GG -3′). The PCR amplified tandem-dimer-Tomato (TdTomato) coding sequence was cloned in Bsp120I-XbaI of pBluescriptSK+. The three polyA signals and Lox2272-kanamycin-Lox2272 cassettes were subsequently added downstream of TdTomato. For SLIC, 5′- CCG CAG CCA TGA GTA TGG TGA GCA AGG GCG AGG AG -3′ and 5′- GCT CCT TTT CAG ACA CGA CTC ACT ATA GGG CGA ATT G -3′ primers were used and pGEM-T-Easy-Myh1 linearized with HindIII. The targeting vector pGEM-T-EasyMyh4CFP was generated by SLIC of a CFP-3pA-LoxN-KanamycinLoxN PCR fragment which will later interfere with our strategy of recombination, BAC DNA amplified in DH10b is extracted (Nucleobond MIDI XTRA, Macherey-Nagel), checked by Acc65I-NotI complex restriction profile, and transformed by electroporation into SW105 competent cells. BAC DNA from several transformants is extracted and checked using the same complex restriction profile against the parental one. Removal of LoxP is carried out on one bacterial clone made competent then induced for recombinase expression by 15 min incubation at 42 °C by electroporation of a 1.85-kb BamHI-NotI DNA fragment purified from pTamp-BACe3.6 (gift of Dr V. Besson) conferring ampicillin resistance. BAC DNA from recombinant ampicillin-resistant clones is extracted and checked against parental DNA using Acc65-NotI or MfeI-NotI complex restriction profiling. Similarly, removal of Lox511 is performed on one ampicillin-resistant clone using a 2.2-kb KpnI-BamHI fragment purified from pSKTHygroBACe3.6Lox511 (gift of Dr J. Hadchouel) which confers hygromycin resistance to recombinant clones. DNA from one clone is then transformed into SW106 cells harboring Cre-inducible expression under arabinose treatment[81] for further targeting step.

Sequential Myh2, Myh4, and Myh1 locus modifications are performed by three rounds of competent bacterial clone electroporation using a 3.75-kb NotI transgene purified from each respective pGEMTe-based targeting vector described above followed by kanamycin selection of recombinant clones, BAC DNA extraction, complex restriction profiling against parental DNA, then from a proper recombinant clone floxing-out kanamycin resistance by 0.1% arabinose treatment, BAC DNA extraction and again complex restriction profiling against parental DNA. Enzymes combinations are as follows: KpnI+NotI and MfeI+NotI for Myh2-YFP, Myh4-CFP-kana, and Myh1-TdT; MfeI+NotI and BamHI+NotI for Myh4-CFP. The final transgenic BAC DNA is then transferred back to DH10b cells for better extraction yield (Nucleobond BAC100, Macherey-Nagel). DNA is resuspended in 10 mM Tris-HCL pH 7.0, 1 mM EDTA, 100 mM NaCl. The final transgenic BAC DNA is then transferred back to DH10b cells for better extraction yield (Nucleobond BAC100, Macherey-Nagel). DNA is resuspended in injection buffer (10 mM Tris-HCL pH 7.0, 1 mM EDTA, 100 mM NaCl), and 200 ng filtrated through drop dialysis against the filtration buffer for 1 h using Millipore cellulose ester disc membranes VMWP 0.05 µm (Ref# VMWP02500). The transgenic mice having integrated the BAC were genotyped using primers amplifying the regions between Myh4 and CFP (forward: 5′- CTG AGC TGC CAC CAA TAG CC, reverse: 5′- CTT GTA GTT GCC GTC GTC CTT). BAC copy number and integrated DNA regions were determined by qPCR on genomic DNA of Enh+ and Enh- transgenic mice with primers along the BAC.

**Fetuses preparation**. Fetuses were staged, taking the appearance of the vaginal plug as embryonic day (E) 0.5, harvested at 18.5 days post fertilization, decapitated and their skin was removed. They were fixed in 4% PFA o/n at +4 °C and kept in 15% sucrose-PBS at +4 °C overnight. Then they were embedded into OCT and snap frozen in isopentane (−30 °C) cooled in liquid nitrogen and kept at −80 °C until used. Transversal trunk 10 µm cryostat sections were thaw-mounted onto poly-L-lysine coated glass slides (Superfrost Plus) and kept until use at −80 °C.

**Immunohistochemistry**. Fetuses sections were rehydrated in PBS before antigene retrieval treatment in a pH6 citrate buffer solution at 95 °C for 15 min followed by 20 min cooling. They were permeabilized and blocked in PBS with 0.5% triton X100 and 10% normal goat serum for 3 h at room temperature. Primary and secondary antibodies were diluted in the blocking solution and incubated on the sections at room temperature overnight and 1 h respectively. Immuno-stained

sections were mounted under a coverslip with Mowiol fluorescent mounting medium before imaging. Images were taken on an Olympus BX63 upright fluorescent microscope, or on a Yokogawa CSU X1 Spinning Disk coupled with a DMI6000B Leica inverted microscope and acquisitions were made with an ORCA-Flash4.0 LT Hamamatsu camera or a CoolSnapHQ2 camera (Photometrics) respectively, with Metamorph 7 software. Immunostaining against YFP and MYH2 were performed on soleus and immunostaining against Tomato and Myh1 were performed on quadriceps. Adult muscles were fixed 30 min in PFA 2% with 0,2% Triton at 4 °C. After overnight 10% sucrose treatment, muscles were embedded with TissuTEK OCT (Sakura) and frozen in cold isopentane cooled in liquid nitrogen. For immunostaining against MYH4, MYH2, MYH7, and Laminin, freshly dissected adult legs without skin were embedded with TissuTEK OCT and directly frozen in cold isopentane cooled in liquid nitrogen Muscles were conserved at −80 °C and cut with Leica cryostat 3050 s with a thickness of 10 μm. Cryostat sections were washed three times 5 min with PBS and then incubated with blocking solution (PBS and 10% goat serum) 30 min at room temperature. Sections were incubated overnight with primary antibody solution at +4 °C, then washed three times for 5 min with PBS and incubated with secondary antibody solution 1 h at room temperature. Sections were further washed three times 5 min and mounted with mowiol solution and a glass coverslip. Images were collected with an Olympus BX63F microscope and a Hamamatsu ORCA-Flash 4.0 camera. Images were analyzed with ImageJ program. The references of the antibodies used are listed in Table S1.

**RNA extraction and quantification**. RNA extractions from adult skeletal muscles were performed using TRIzol reagent (ThermoFischer) following the manufacturer's protocol. Muscles were lysed with Tissue lyser (Quiagen) in TRIzol solution. RNA was precipitated with isopropanol. cDNA synthesis was performed with Superscript III kit (Invitrogen) using 1 μg of RNA. RT-qPCR were performed using Light Cycler 480 (Roche) with the Light Cycler 480 SYBR Green I Master Kit (Roche) following the manufacturer's protocol with 40 cycles at 95 °C for 15 s, 60 °C for 15 s, and 72 °C for 15 s. We used 36B4 housekeeping gene to normalize the expression level between different samples. The sequences of the oligonucleotides used are listed in Table S2.

**Single-nucleus ATAC-seq from skeletal muscle**. We use the 10X genomic nuclei Isolation for Single Cell ATAC Sequencing protocol (CG000169 | Rev B) with some changes. 12 quadriceps and 12 soleus were dissected and pulled in cold PBS. PBS was removed and muscles were minced 2 min in 1 ml of cold ATAC-lysis buffer (10 mM Tris-HCl pH 7.4, 10 mM NaCl, 3 mM MgCl2, 1% BSA, and 0.1% Tween-20 in Nuclease-Free Water). In all, 6 ml of cold ATAC-lysis buffer were added and muscles were lysed on ice. After 3 min the lysate was dounced with 10 strokes of loose pestle avoiding too much pressure and air bubbles. After douncing, 8 ml of wash buffer were added and the homogenate was filtered with 70 μm, 40 μm, and 20 μm cell strainers. Nuclei were pelleted by centrifugation for 5 min at $500 \times g$ at +4 °C. Next, we used the Chromium Single Cell ATAC kit according to the manufacturer's protocol. Nuclei were resuspended in nuclei buffer from the kit, transposed 1 h at 37 °C. We loaded around 6000 nuclei into the 10X Chromium Chip. GEM incubation and amplification were performed in a thermal cycler: 72 °C for 5 min, 98 °C for 30 s and 12 repeated cycles of 98 °C for 10 s, 59 °C for 30 s, and 72 °C for 1 min. Post GEM Cleanup using DynaBeads MyOne Silane Beads was followed by library construction (98 °C for 45 s, cycled 12 × 98 °C for 20 s, 67 °C for 30 s, and 72 °C for 1 min). The libraries were constructed by adding sample index PCR, and SPRIselect size selection. The fragment size estimation of the resulting libraries was assessed with High SensitivityTM HS DNA kit runed on 2100 Bioanalyzer (Agilent) and quantified using the QubitTM dsDNA High Sensitivity HS assay (ThermoFisher Scientific). Libraries were then sequenced by pair with a HighOutput flowcel using an Illumina Nextseq 500.

**Single-nucleus ATAC-seq analysis**. A minimum of 10 000 reads per nucleus were sequenced and analyzed with Cell Ranger Single Cell Software Suite 3.0.2 by 10X Genomics. Raw base call files from the Nextseq 500 were demultiplexed with the cellranger-atac mkfastq pipeline into library-specific FASTQ files. The FASTQ files for each library were then processed independently with the cellranger count pipeline. This pipeline used STAR21 to align reads to the *Mus musculus* genome. Once aligned, barcodes associated with these reads –cell identifiers and Unique Molecular Identifiers (UMIs), underwent filtering and correction. The subsequent visualizations, clustering and differential expression tests were performed in R (v 3.4.3) using Seurat36 (v3.0.2)[82], Signac (v0.2.4) (https://github.com/timoast/signac) and Chromvar (v1.1.1)[83]. Quality control on aligned and counted reads was performed keeping cells with peak_region_fragments >3000 reads and <100,000, pct reads in peaks >15, blacklist ratio <0.025, nucleosome_signal <10 and TSS.enrichment >2. We get 6037 nuclei in total and we detected 132,966 peaks. The pseudo-bulk accessibility tracks of the *fMyh* locus were generated with the coverage plots function of Signac. The myonuclei expressing the different isoforms of *Myh* were classified according to the opening of the chromatin at the level of the promoters of the different genes of *Myh*. The number of nuclei used for this analysis was 64 for *Myh7*, 59 for *Myh2*, 249 for *Myh1*, and 495 for *Myh4*. Single-nucleus ATAC-seq tracks were visualized using IGV software version 2.3.70.

**ChIP-seq analysis**. Fastq files of quadriceps femoris and soleus H3K27ac ChIP-seq[25] were download from the GEO database (accession number GSE123879). Fastq files of quadriceps femoris Mll4 ChIP-seq[26] (accession number GSE137285) were download from the GEO database. Fastq files of CTCF ChIP-seq[84] (accession number GSE138994) were download from the GEO database.The reads were aligned to the mouse mm10 genome using bowtie2[85] and peaks were called by MACS2[86] using q value cutoff = 0.05. ROSE algorithm[14] was applied to identify and rank the enhancers based on H3K27ac ChIP-seq signal, with a stitching distance of 12.5 kb. Chip-seq tracks were visualized using IGV software version 2.3.70.

**Nuclei purification from adult skeletal muscle for 4C-seq**. Nuclei purification from adult skeletal muscle has been performed as previously described[87] with some modifications. After dissection, 16 soleus or 8 quadriceps were resuspended in 1 mL of hypotonic buffer (25 mM Hepes-KOH pH 7.8, 10 mM KCL, 1.5 mM MgCl2, 0.1% NP40, PIC 1X (complete protease inhibitor Roche), PMSF 1 mM) in a 2 ml tube for 5 min at +4 °C. Muscles were sliced with a scissor for 30 sec. The small pieces of muscles were transferred into a round tube of 15 mL at +4 °C and 4 ml of cold hypotonic buffer was added. After 5 min the solution was homogenized for 15 s with an Ultra-Turrax (IKA) at a speed of 17,500 rpm. The solution was transferred in a 15 ml Falcon tube and crosslinked with 2% formaldehyde (in a volume of 10 ml of hypotonic buffer) at room temperature during 10 min. In all, 1.43 ml of cold glycine (1 M) was added to quench the formaldehyde for 5 min at +4 °C while shaking. The crosslinked nuclei were dounced 10 times with a loose pestle and then centrifuged at 1000×g for 10 min at +4 °C. The nuclear pellet was resuspended in 5 ml of hypotonic buffer and filtered with 70 μm and 40 μm cells strainers. The nuclei were pelleted with centrifugation at 1000×g for 10 min, snap frozen into liquid nitrogen and stored at −80 °C.

**4C-seq**. 4C-seq experiments have been performed as previously described[88] with some modifications. Purified crosslinked nuclei from 160 soleus and 80 quadriceps were pooled together to have 10⁷ nuclei per condition. PCR primers were designed for each viewpoint according to the protocol. The first digestion was done with DpnII (New England Biolabs) and the second with NlaIII (New England Biolabs). For each viewpoint 800 ng of 4C template was amplified by PCR. The samples were sequenced on the Illumina NextSeq 500 platform, using 75 bp single end reads. The analysis of the data has been done using the HTSstation 4C-seq pipeline[89]. Briefly, sequences were demultiplexed, aligned to the reference genome (mm10), translated back to DpnII restriction fragments and normalized. For visualizations the fragments 2 kb up- and downstream of the viewpoint were excluded, followed by smoothing of 4C-seq signal (11 fragments running mean) and normalization to the five TADs surrounding the fMyh TAD (chr11: 66,613,299–68,004,316), following a previously published strategy[90]. Ratios between smoothened 4C-seq patterns were calculated using the BioScript library of the HTS station[89]. Significance of difference in the distribution of the 4C-seq signal was calculated using a previously applied approach[91] by normalizing the unprocessed 4C-seq signal within the above-mentioned five TADs followed by determining the fraction of fragments with increased and decreased 4C-seq signal in the quadriceps versus the soleus in the region spanning the fMyh SE (chr11: 67,103,534–67,145,377) versus the remainder of the fMyh TAD (chr11: 67,099,993–67,349,955) followed by a G test of independence.

Hi-C data in mouse ES cells were obtained from the 3D Genome Browser website (http://promoter.bx.psu.edu/hi-c/view.php). ChIP-seq data against CTCF in mouse ES cells and DNase I hypersensitive site in adult fast skeletal muscle were obtained from the ENCODE database. The sequences of the oligonucleotides used for 4C-seq are listed in Table S3.

**Mouse generation by CRISPR/Cas9**. SgRNA and Cas9 purified protein were produced by the TACGENE platform. The SgRNA was designed with the Crispor program (http://crispor.tefor.net/)[92]. SgRNA is produced by T7 Hiscribe transcription kit (New England Biolabs) and purified by EZNA microelute RNA clean up kit (Omega biotek). The DNA used for transcription was produced by overlapping PCR. For each cut sites, 3 different sgRNA were designed and tested in vitro by transfection in MEF cells. The deletions were performed by injecting into oocytes between 1 and 5 pg of sgRNA (60 ng/μl) cutting at both sides of the deletion and of the Cas9 protein (30 μM). Oocytes were reimplanted into a pseudopregnant females. Mutant mice were screened by PCR and confirmed by sequencing. The list of the sgRNA and PCR primers used for screening are listed in Table S4.

**FISH with amplification (RNAscope) on isolated fibers**. RNAscope® Multiplex Fluorescent Assay V2 was used to visualize fast *Myh* pre-mRNAs and mRNAs. Twenty different pairs of probes against the first intron of each fast *Myh* transcript were designed by ACDbio. Muscles were dissected and immediately fixed in 4% PFA at +4 °C during 30 min. After fixation muscles were washed three times in PBS for 5 min. Myofibers were dissociated mechanically with small tweezers and fixed onto Superfrost plus slides (Thermo Fischer) coated with Cell-Tak (Corning) by dehydration at +55 °C during 5 min. Slides were then proceeded according to the manufacturer's protocol: ethanol dehydration, 10 min of H2O2 treatment and 30 min of protease IV treatment. After hybridization and revelation, the fibers were mounted under a glass coverslip with Prolong Gold Antifade Mountant

(Thermofischer). Myofibers were imaged with a Leica DMI6000 confocal microscope composed by an Okogawa CSU-X1M1 spinning disk and a CoolSnap HQ2 Photometrics camera. Images were analyzed with Fiji Cell counter program.

**Statistical analysis**. The graphs represent mean values ±SEM. Significant differences between mean values were evaluated using two-way ANOVA for Fig. 2H, one-way ANOVA with multiple comparisons for Figs. 3D, 4E, and 5C–E, and Sup 8E and 9F–H and Student's *t* test for Figs. 1B and 2K with Graphpad 8.4.3 software. Immunostaining and FISH experiments were repeated 3 times independently with similar results. SnATAC-seq experiments were repeated two times independently with similar results.

**Reporting summary**. Further information on research design is available in the Nature Research Reporting Summary linked to this article.

## Data availability

The authors declare that all data supporting the findings of this study are available within the article and its supplementary information files or from the corresponding author upon reasonable request. All 4C-seq data are available in the NCBI Gene Expression Omnibus (GEO) database under accession number "GSE168074". All snATAC-seq data are available in the NCBI Gene Expression Omnibus (GEO) database under accession number "GSE150065". Source data are provided with this paper.

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

## Acknowledgements

We thank D. Duboule, S. Schiaffino, H. Amthor, M. Schuelke, E. Bloch-Gallego, and F. Spitz for helpful discussions. We thank U. Schibler, S. Gautron, and F. Britto for helpful critical reading of the manuscript. We thank T. Guilbert and F. Letourneur at the Cochin IMAG'IC and GENOM'IC platform for helpful advice. We acknowledge the High-throughput sequencing facility of the I2BC (Gif-sur-Yvette, France) for its sequencing and bioinformatics expertize. M.D.S was supported successively by a PhD fellowship from the Association Française contre les Myopathies (AFM), by a post doc fellowship from EUR-Gene and from the Agence Nationale pour la Recherche (ANR MYOLINC). Financial support to this work was provided by the AFM (no. 16427, 21711, and 23012), the ANR (Myocodes -09-BLAN-0280-01 and MYOLINC -16-CE14-0032-01), the Institut National de la Santé et de la Recherche Médicale (INSERM), and the Centre National de la Recherche Scientifique (CNRS) to P.M., by AFM (no. 21711) to J.D., and by AFM (no. 19507 and 22946 Translamuscle) to F.R.

## Author contributions

Designed experiments: M.D.S., D.N., I.S., F.A., S.B. and P.M. Performed experiments: M.D.S., I.S., S.B., M.Wu., F.A., R.P., F.L., M.D.C., A.Sc., A.So., and J.-P.C. Interpreted data: M.D.S., I.S., D.N., S.B., A.Sc., A.So. and P.M. Bioinformatic analysis: M.Wo. and M.D.S. Wrote the manuscript: M.D.S. and P.M. with input from D.N., A.So., and J.D. Funding acquisition, P.M., J.D. and F.R.

## Competing interests

The authors declare no competing interests.
