## [Peer Review File · Nature Communications]

Reviewers' Comments:

Reviewer #1:

Remarks to the Author:

I read with great interest the manuscript by Dos Santos et al. entitled "A fast Myh super enhancer dictates adult muscle fiber phenotype through competitive interactions with the fast Myh genes". In this manuscript the authors use single-nucleus ATAC-seq in order to identify a super-enhancer (SE) or LCR that regulates the fast Myh genes in muscle cells. Then, using 4C-seq analyses and genetic engineering experiments in mice they demonstrate the functional relevance of the SE and provide insights into the mechanism by which the SE can specifically interact with a unique Myh gene in each muscle cell. Overall, this is a very thorough, high-quality study that provides important insights into the regulatory function of SE/LCR and the mechanisms whereby gene clusters can be dynamically and specifically regulated. Although I overall support publication in Nature Communications, there are a number of issues that I think the authors should try to improve:

Major comments:

1. My main conceptual concern with the presented work is that the authors proposed that the SE controls the specific expression of a single fMyh gene in different muscle cells through "exclusive interactions with a single promoter" at the Myh locus. However, these conclusions are based on 4C-seq experiments that are performed on cell populations that do not allow the authors to distinguish exclusive from multi-way interactions or to infer 3D topologies at the single cell level. In fact in Fig 1H, one can appreciate that when using the Myh4 promoter as a viewpoint, contacts can be observed not only with the SE but also with the Myh1, Myh2 and Myh8 promoters in the quadriceps. Similarly, when using the Myh2 promoter as a viewpoint, contacts with the SE and the Myh4 promoter can be seen in the soleus. The interactions between the SE and several Myh promoters would support a hub model rather than exclusive interactions between the SE and a single Myh gene promoter. How do the authors explain these results?. Therefore, the authors should consider toning down their claims, being more speculative, or, alternatively, they could perform additional experiments using imaging (high resolution DNA-FISH) or single-allele 3C methods to support their claims.
2. The authors should comment not only in the downregulation of Myh4 but also on the increased expression of Myh2 in the EnhA^{-/-} mice. How can the authors explain the Myh2 upregulation?. In Fig 1D one can appreciate that the ATAC-seq signal for the EnhA region is strong in Myh4 cells but rather weak in Myh2 cells. Could the EnhA region act as a silencer in Myh2 cells?.
3. Based on the deletions and inversions generated in the fMyh locus the authors conclude that the different fMyh genes compete for the SE. However, I don't think that the presented data fully supports this conclusion and alternative models might be still possible.
 - In the Myh(1-4)Del/Del mice the changes in Myh2 expression are rather mild and not statistically significant, at least based on qPCR analyses (Fig 5E), which is not concordant with what the authors write in the text and with the proposed competition model.
 - In the Myh(1-4)Del/Del mice, the increased expression of Myh8 and Myh13 could be due to a distance effect and/or due to the elimination of some insulator/boundary. For example, in Fig S2B there seems to be a CTCF peak just upstream of Myh4 that could act as a boundary that prevents the communications between the SE and Myh8/13. Moreover, in the two different Myh(1-4) inversion mouse models, the authors observe increased expression of Myh8 and Myh13, although at much lower levels than in Myh(1-4)Del/Del mice (Fig 5E-G; please use same scales across the different mouse models for the same genes). This could argue that the reduced distance between the SE and the Myh8/13 genes in the deletion mice has a major effect. The authors could introduce a small deletion that removes the Myh4 promoter (without disrupting the near CTCF peak) without significantly affecting the distance separating the SE and Myh8/13 genes. Alternatively, to add further support to promoter competition as a mechanism to explain how the SE controls specific Myh genes, an inversion between Myh1 and Myh13 could be quite useful.
 - I acknowledge that generating new mouse models is a time consuming and demanding task and the authors have already generated an impressive set of transgenic mouse lines. However, they should be more critical when evaluating the presented data and should be opened to consider other models beyond promoter competition.

Minor comments:

1. The computational analyses used to classify the nuclei based on the snATAC-seq signals for the different Myh genes should be more extensively described in the Methods section. How many nuclei are being pooled for each "Myh gene" category in Fig 1D?
2. The loss of linc-Myh expression in the EnhB^{-/-} mice should be emphasized, as it shows that the SE rather than the lincRNA are functionally relevant with respect to fMyh expression control.

Reviewer #3:

Remarks to the Author:

In the article of Dos Santos entitled "A fast Myh super enhancer dictates adult muscle fiber phenotype through competitive interactions with the fast Myh genes", the authors describe a 42 kb super-enhancer at the fast Myh locus. Combining various cutting-edge technics including snATAC-seq, 4C-seq, and CRISPR/Cas9 with a rainbow transgenic mouse model recapitulating the endogenous spatio-temporal expression of adult fMyh genes, they reveal an active competition of the promoters for this super-enhancer.

This promising study requires however various controls and writing adjustments prior to publication.

Major points.

Since the paper is mainly based on omics data, results from the peaks calling should be shown for the various experiments. For instance, authors mention that they observed 7 chromatin accessibility peaks in an intergenic region between Myh3 and Myh2 (Figure 1D) without showing the peak locations.

Figure 1 and associated supplementary figures.

As, super enhancers are generally characterized by the presence of various factors, including Med1, the authors should provide ChIPseq data for some of these factors.

The software that was used for visualization should be given, as well as the scale for the read intensity.

The information depicted in Figures 1F and 1G should be more explicit.

For 4C-seq experiments, please clearly show the viewpoint location and provide information on the significance of the contact domains in Figure 1H.

Please explain the data presented in Figure S1B.

Figure S2B is presented prior to S2A.

Figure S2A is mentioned as Figure S 2 in the main text.

The ChIP-seq used for Ctf does not seem to provide significant peaks (Figure S2B). Please use another data set and show results from peak calling.

Viewpoints in Figure S2B should be clearly indicated.

Figure 2 and associated supplementary figures.

Please label Figure S5A as S5, since there is no S5B.

The authors should further comment the various conditions depicted in Figure S6.

Figure 3 and associated supplementary figures.

The basic histology of limbs, ribs and diaphragm of control and SE ^{-/-} mice should be provided. Figure 3G is presented before 3D.

Figures 3E and F are presented before 3D.

Page 9: "At the limb level expression of Myh3 and Myh7 was not affected as shown by RT-qPCR experiments and by immunocytochemistry against MYH3 and MYH7 (Figures 3G, H, S7C)." There are no RT-qPCR experiments in these figures.

Figures 3J and 3K are presented before 3I.

Please state on what is based the assumption that the "defects of sarcomere formation in mutant myofibers did not impair their innervation but seemed to affect neuromuscular junctions

distribution in the diaphragm (Figure 3I)".

Figure 4 and associated supplementary figures.

The authors should explain how the 2 CRMs were defined in the result section.

Transcription factor are in general recruited to DNA in a valley of H3K27ac. Have the authors performed a motif search at these particular position to determine key factors controlling fMyh expression?

Figure 5 and associated supplementary figures.

The authors should clarify by which mechanism the various promoters can compete for the described SE. Is it due to a myofiber-specific set of transcription factors?

Minor points

Experiments were performed in females. Are the results similar in males ?

English and writing should be improved.

Page 11: The homozygote Myh(1-4)Inv/Inv mutant mice were viable at the homozygous state should be : The homozygote Myh(1-4)Inv/Inv mutant mice were viable.

Page 12 : This showing... should be : This showed...

Page 16: The human body is composed of more than 600 different skeletal muscles, each with specific functions and properties.

Is redundant with Page 12: At least 640 different skeletal muscles can be identified in the human body, each with a specific form, architecture, position, and function.

The authors should be more straightforward in their conclusions, and not only write that "their results suggest".

Page 9 : Dos Santos should be ref 10.

Page 24 and figure legends: data are presented as mean values +/- SEM, whereas in the figures it is only + SEM.

The significance of the sequences provided in Fig. S7B, S8B and C and S9B and D should be clarified.

The reference of alpha bungarotoxin in table S1 is not an antibody reference.

The sequence of the primers in Table S2 should be homogenized.

Reviewer #4:

Remarks to the Author:

This manuscript reports the discovery of a super enhancer that controls expression of myosin genes in skeletal muscle. They use snATAC-seq and ChIP-seq to identify a large region of open chromatin in the myosin clusters, then generate mouse models to assess sufficiency and necessity of the super enhancer. Overall, the results are important and interesting for the field as it begins to tease apart the mechanisms by which myosin genes are regulated to control muscle fiber type and function. The majority of the claims are supported by the data but there are some instances of unclear interpretations, which should be adjusted prior to publication. The questions related to technical issues or controls are mainly minor. Specific comments are below.

1. Genotyping for the Enh- and Enh+ should be shown, especially related to the comment that

'Enh+ integrated 2 complete copies'. How many copies does the Enh- line contain?

2. The meaning for the results from the experiments deleting two independent cis-regulatory module within the super enhancer are not clear. The authors say that there are distinct enhancer elements that possess distinct functions and then show a schematic at the end of the figure. Do the data support the idea that Myh8 and Myh13 are down since the data are not significant? How is there a reduction of Myh1 protein in 4D but no change in mRNA levels in 4E? The schematic at the end is also confusing. Specifically, there is no connection between enhancer B and Myh4 even though Myh4 is increased. Is there a way to simplify this model?

3. I am not sure if the data in Fig. 5 definitively show that the myosin gene promoters 'compete' for the super enhancer. Some manipulations result in deletions and thus fewer promoters, whereas others are inversions, so availability of promoters and spacing has been altered. Maybe 3D chromatin looping experiments would show competition.

4. The arbitrary classification of muscles into three categories is not clear. I think the issue is that the logic for this is not explained.

5. The idea that there are differences in regulation based on location within the muscle in one of the mutants (Myh2 upregulated in deep regions) could use more data for support. Were multiple planes of section analyzed to know if the Myh2 is regional in terms of the length of the fiber? Some quantification for this across animals would also be helpful?

6. It is mentioned in the abstract that the work could explain how some regions of muscles or certain muscles are spared or more impacted in muscle diseases. This is a bit of an over-interpretation as there is not any presented that would directly support such an argument. Alternatively, perhaps the authors could more clearly explain their reasoning for the interpretation.

REVIEWER COMMENTS

Reviewer #1 (Remarks to the Author):

I read with great interest the manuscript by Dos Santos et al. entitled “A fast Myh super enhancer dictates adult muscle fiber phenotype through competitive interactions with the fast Myh genes”. In this manuscript the authors use single-nucleus ATAC-seq in order to identify a super-enhancer (SE) or LCR that regulates the fast Myh genes in muscle cells. Then, using 4C-seq analyses and genetic engineering experiments in mice they demonstrate the functional relevance of the SE and provide insights into the mechanism by which the SE can specifically interact with a unique Myh gene in each muscle cell. Overall, this is a very thorough, high-quality study that provides important insights into the regulatory function of SE/LCR and the mechanisms whereby gene clusters can be dynamically and specifically regulated. Although I overall support publication in Nature Communications, there are a number of issues that I think the authors should try to improve:

We thank the reviewer for his/her constructive remarks.

Major comments:

1. My main conceptual concern with the presented work is that the authors proposed that the SE controls the specific expression of a single fMyh gene in different muscle cells through “exclusive interactions with a single promoter” at the Myh locus.

However, these conclusions are based on 4C-seq experiments that are performed on cell populations that do not allow the authors to distinguish exclusive from multi-way interactions or to infer 3D topologies at the single cell level. In fact in Fig 1H, one can appreciate that when using the Myh4 promoter as a viewpoint, contacts can be observed not only with the SE but also with the Myh1, Myh2 and Myh8 promoters in the quadriceps. Similarly, when using the Myh2 promoter as a viewpoint, contacts with the SE and the Myh4 promoter can be seen in the soleus. The interactions between the SE and several Myh promoters would support a hub model rather than exclusive interactions between the SE and a single Myh gene promoter. How do the authors explain these results? Therefore, the authors should consider toning down their claims, being more speculative, or, alternatively, they could perform additional experiments using imaging (high resolution DNA-FISH) or single-allele 3C methods to support their claims.

We agree with the reviewer that these observations, and particularly the minor yet consistently noticeable interactions among the Myh-promoters should have been discussed and interpreted in a more careful context.

To get a better appreciation of the cell type-specific dynamics of these interactions, we have calculated for the Myh2 and Myh4 viewpoints the ratios between the cell types, including the statistical relevance of these differences (Fig. 1H, taking approaches as first detailed in Noordermeer et al, eLife 2014 and Thierion et al, Plos Genetics 2018). This analysis confirms that the activity of Myh4 and Myh2 is directly associated with increased interactions with the Super Enhancer, and that these increases are highly significant as compared to the reorganization of 3D contacts in the remainder of the TAD.

A less defined pattern emerges for other regions in the TAD, particularly the promoters. In their inactive cell type, the viewpoints (Myh4 in Soleus and Myh2 in Quadriceps) display a minor increase in local contacts that spreads out over 50 – 100 kb, suggesting they adopt a more compacted organization. In the case of Myh2 in the Quadriceps, this specifically includes the Myh1 promoter and gene body as well, but not the Myh4 gene. In contrast, the active Myh4 viewpoint in the Quadriceps engages in moderate yet clearly apparent interactions with the Myh1 and Myh2 genes. As such, no clear model emerges, but rather we may be looking at different cell populations (e.g. cell populations in the Quadriceps where Myh4 is active or inactive), with the active gene decompacting and looping out of a more compacted hub. Depending on distance and activity state, this may then give different interaction patterns.

Based on the literature it seems that ability of promoters to be engaged in a liquid phase separation droplet with a super enhancer depends on a high concentration of bound transcription factors and cofactors, what is not the case for Myh8, Myh1 or Myh2 promoters in Myh4+ myonuclei as revealed by snATAC-seq experiments. We agree that a low amount of bound transcription on these promoters in Myh4+ myonuclei could be under the detection threshold for this experiment, but once again this low

undetected amount of TF would argue against the participation of these promoters in a hub model with the SE.

Based on the outcome of our analysis, we have changed the discussion to discuss alternative scenarios, including the hub model that, as we discuss, we do not favor but cannot exclude.

We appreciate the reviewer's suggestion for super-resolution DNA-FISH (oligo-paints, ORCA) or single-allele 3C studies and agree that such studies in the future may provide additional detail on the 3D organization of this locus. Such studies are highly time consuming and require both specialized personnel and considerable funds. At this stage we therefore think it goes beyond the scope of the manuscript.

2. The authors should comment not only in the downregulation of Myh4 but also on the increased expression of Myh2 in the EnhA^{-/-} mice. How can the authors explain the Myh2 upregulation?. In Fig 1D one can appreciate that the ATAC-seq signal for the EnhA region is strong in Myh4 cells but rather weak in Myh2 cells. Could the EnhA region act as a silencer in Myh2 cells?

This is a valid comment, as the activation of Myh2 in EnhA^{-/-} cells is convincing, and we observed an increased number of MYH2⁺ myofibers in the TA. As discussed for the new figure 1H, the 4C-seq interactions of Myh2 with Enhancer A are only very mildly increased in Sol where Myh2 is expressed over Quad where it is not. So we do not detect increased expression of Myh2 in Myh2⁺ myonuclei of the Sol, arguing that in this muscle element A is not bound by negative elements. EnhA may nevertheless act as a silencer of Myh2 in Myh4⁺ myonuclei. Additive transgenesis would allow to test the impact on the A sequence upon Myh2 (and Myh1) promoter sequence regulation and would allow to conclude on the possibility, that we now discuss, that A element is bound by TF that antagonize Myh2 expression, (and mainly Myh1 expression in peroneal muscles as observed by immunohistochemistry). In the competition model that we propose decreased specific TF binding to the SE in Myh4⁺ myonuclei (of the TA) due to the absence of element A weakens SE-Myh4-promoter interactions and may favor the balance to SE-Myh2 interactions (or SE-Myh1 interactions depending on the muscle).

3. Based on the deletions and inversions generated in the fMyh locus the authors conclude that the different fMyh genes compete for the SE. However, I don't think that the presented data fully supports this conclusion and alternative models might be still possible.

- In the Myh(1-4)Del/Del mice the changes in Myh2 expression are rather mild and not statistically significant, at least based on qPCR analyses (Fig 5E), which is not concordant with what the authors write in the text and with the proposed competition model.

In this model where Myh1 and Myh4 promoters are absent we observed strong upregulation of Myh8 and Myh13, but not of Myh2. We better explain that there is a strong competition between Myh4/Myh1 and Myh8. During fetal development Myh8 is activated in all fetal myonuclei and absence of the SE leads to its inhibition. During post natal development there is a switch between Myh8 and either Myh2, Myh1 or Myh4 leading to the down regulation of Myh8 for the benefit of adult Myh expression. If Myh4 promoter is absent we suspect that Myh8 remains expressed in those fibers that should express Myh4 and all the genes associated with Myh4 expression since the TF machinery should not have been changed in those fibers programmed to be Myh4⁺. So we think that the competition model is a good explanation, but cannot exclude another explanation. Both are discussed.

- In the Myh(1-4)Del/Del mice, the increased expression of Myh8 and Myh13 could be due to a distance effect and/or due to the elimination of some insulator/boundary. For example, in Fig S2B there seems to be a CTCF peak just upstream of Myh4 that could act as a boundary that prevents the communications between the SE and Myh8/13. Moreover, in the two different Myh(1-4) inversion mouse models, the authors observe increased expression of Myh8 and Myh13, although at much lower levels than in Myh(1-4)Del/Del mice (Fig 5E-G; please use same scales across the different mouse models for the same genes). This could argue that the reduced distance between the SE and the Myh8/13 genes in the deletion mice has a major effect. The authors could introduce a small deletion that removes the Myh4 promoter (without disrupting the near CTCF peak) without significantly affecting the distance separating the SE and Myh8/13 genes. Alternatively, to add further support to promoter competition as a mechanism to explain how the SE controls specific Myh genes, an inversion between Myh1 and Myh13 could be quite useful.

We agree with the reviewer comments and cannot definitively exclude his/her arguments concerning the involvement of the distance. Nevertheless as explained in the manuscript in the Myh(1-4)inv, Myh1 is no more expressed. Absence of Myh1 expression (the distance between the SE and Myh8 being unchanged) leads to Myh8 upregulation probably because Myh1 promoter is no more able to compete

and to be actively incorporated in the phase separation droplet created by the SE. Myh13 upregulation is observed in Myh(1-4)inv3' when MYH4 protein is no more synthesized, but to a lower level than observed after the deletion of Myh1 and Myh4 (Myh(1-4)Del).

We modified the scales in Figure 5.

• I acknowledge that generating new mouse models is a time consuming and demanding task and the authors have already generated an impressive set of transgenic mouse lines. However, they should be more critical when evaluating the presented data and should be opened to consider other models beyond promoter competition.

Although we still think that promoter competition is the most likely mechanism, the reviewer is right that there is no definitive proof for this claim. So we toned down our conclusions and discuss the two models.

Regarding CTCF: the reviewer is right about a peak on the Myh4 promoter, but it is not very prominent. Both the Hi-C and 4C-seq results do not give the impression that it has a strong insulating function. So we have no evidence that this CTCF peak would for instance function to create a sub-division / sub-TAD.

Minor comments:

1. The computational analyses used to classify the nuclei based on the snATAC-seq signals for the different Myh genes should be more extensively described in the Methods section. How many nuclei are being pooled for each "Myh gene" category in Fig 1D?.

We have now included the required informations in the Methods section.

2. The loss of linc-Myh expression in the EnhB^{-/-} mice should be emphasized, as it shows that the SE rather than the lincRNA are functionally relevant with respect to fMyh expression control.

We agree with the reviewer and now emphasize the results obtained in EnhB^{-/-} animals that are in agreement with the deletion of the entire Linc-Myh gene corresponding to a 8kb region deleting two distinct snATAC-seq peaks which also has no major impact on MYH1, MYH2 or MYH4 positive myofiber number in the TA or soleus (Schutt et al., 2020) as we now discuss.

Reviewer #2 (Remarks to the Author):

In the article of Dos Santos entitled "A fast Myh super enhancer dictates adult muscle fiber phenotype through competitive interactions with the fast Myh genes", the authors describe a 42 kb super-enhancer at the fast Myh locus. Combining various cutting-edge technics including snATAC-seq, 4C-seq, and CRISPR/Cas9 with a rainbow transgenic mouse model recapitulating the endogenous spatio-temporal expression of adult fMyh genes, they reveal an active competition of the promoters for this super-enhancer.

This promising study requires however various controls and writing adjustments prior to publication.

We thank the reviewer of his/her enthusiasm and for his/her constructive remarks.

Major points.

Since the paper is mainly based on omics data, results from the peaks calling should be shown for the various experiments. For instance, authors mention that they observed 7 chromatin accessibility peaks in an intergenic region between Myh3 and Myh2 (Figure 1D) without showing the peak locations.

We showed the peak locations in Sup1 B in the previous version, and now better explain why we talk about 7 peaks and the rationale between peaks 1-7 and mutants of enhancer A and B. We now present the peaks calling as required.

Figure 1 and associated supplementary figures.

As, super enhancers are generally characterized by the presence of various factors, including Med1, the authors should provide ChIPseq data for some of these factors.

As far as we are aware there is no consensus definition of super enhancers that have been identified. We do not think that Med1 has a major role in the fast Myh SE activity, its absence in skeletal muscles leading to a mild phenotype at the fMyh locus (A muscle-specific knockout implicates nuclear receptor coactivator MED1 in the regulation of glucose and energy metabolism ; Wei Chen et al ; PNAS, 2010, 107, 10196) . In Supp1B, we now present ChiP-seq data from adult skeletal muscle chromatin with Mll4 a methyltransferase enriched in super enhancer.

The software that was used for visualization should be given, as well as the scale for the read intensity.

We added the information in the methods.

The information depicted in Figures 1F and 1G should be more explicit.

We modified the information depicting Figures 1F an 1G in the legends of the figures, and hope that they are now more explicit.

For 4C-seq experiments, please clearly show the viewpoint location and provide information on the significance of the contact domains in Figure 1H.

We have pinpointed the location of viewpoints using arrows in Figure 1H. For a better appreciation of differences in contacts between cell types, we have added ratio tracks. Using a strategy described in detail (Thierion et al, Plos Genetics 2018), we have determined the significance of the differences in contacts (see Fig. 1H and new Fig S2A).

Please explain the data presented in Figure S1B.

The explanations concerning the data presented in Figure S1B have been modified, and we hope that in the present form they are satisfactory.

Figure S2B is presented prior to S2A. Figure S2A is mentioned as Figure S 2 in the main text.

A new panel S2A has been added (relative to Figure 1H). The previous panels S2A and S2B have now been merged into panel S2B, which solves the problem of the order of presentation.

The ChIP-seq used for Ctf does not seem to provide significant peaks (Figure S2B). Please use another data set and show results from peak calling.

We have modified the ChIP-seq data for CTCF.

Viewpoints in Figure S2B should be clearly indicated.

We added arrows to identify the viewpoints.

Figure 2 and associated supplementary figures.

Please label Figure S5A as S5, since there is no S5B.

We modified accordingly.

The authors should further comment the various conditions depicted in Figure S6.

The conditions depicted in Figure S6 have been better commented in the new discussion.

Figure 3 and associated supplementary figures.

The basic histology of limbs, ribs and diaphragm of control and SE -/- mice should be provided.

We added basic HE histology in a new Supplementary figure (Sup7) as required.

Figure 3G is presented before 3D.

We modified accordingly.

Figures 3E and F are presented before 3D.

We modified accordingly.

Page 9: "At the limb level expression of Myh3 and Myh7 was not affected as shown by RT-qPCR experiments and by immunocytochemistry against MYH3 and MYH7 (Figures 3G, H, S7C)." There are no RT-qPCR experiments in these figures.

We apologize for the mistake and modified accordingly.

Figures 3J and 3K are presented before 3I.

We modified accordingly.

Please state on what is based the assumption that the "defects of sarcomere formation in mutant myofibers did not impair their innervation but seemed to affect neuromuscular junctions distribution in the diaphragm (Figure 3I)".

We observed that mutant myofibers of the diaphragm are innervated, they present sarcomere defects. The alpha bungarotoxin and Neurofilament stainings are modified as compared with diaphragm of the control E18.5. We modified the sentence in the text.

Figure 4 and associated supplementary figures.

The authors should explain how the 2 CRMs were defined in the result section.

We better explain how the two CRM A and B were defined by snATAC-seq experiments.

Transcription factor are in general recruited to DNA in a valley of H3K27ac. Have the authors performed a motif search at these particular position to determine key factors controlling fMyh expression?

That is probably try at the level of a few hundred bps, where a hypersensitive site will be flanked by H3K27ac-marked nucleosomes. In this case the valley is much larger (multiple kbs), according to the ATAC-seq, and we have identified multiple SIX, SOX, E box, MEF2 and of many other TF binding sites. Since we have no evidence for the presence of these TF and their binding to the opened regions in adult muscles, except for SIX1, we did not discuss this point that will be addressed in a future study to link motoneuron activity and fMyh gene expression.

Figure 5 and associated supplementary figures.

The authors should clarify by which mechanism the various promoters can compete for the described SE. Is it due to a myofiber-specific set of transcription factors?

See also comment 3 from reviewer 1. No definitive answer can be rigorously addressed, but the hypothesis that myofiber-specific transcription factors accumulation is a logical one (i.e. involved in initiating or stabilizing specific promoter-enhancer loops) and we now better explain this hypothesis in the discussion by introducing SIX homeoproteins that are known to control the expression of the fMyh locus.

Minor points

Experiments were performed in females. Are the results similar in males?

We cannot answer to this question, we did not yet perform experiments in males, but we suspect that the major conclusions drawn in our studies are not sex dependent.

English and writing should be improved.

Page 11: The homozygote Myh(1-4)Inv/Inv mutant mice were viable at the homozygous state should be : The homozygote Myh(1-4)Inv/Inv mutant mice were viable.

Page 12 : This showing... should be : This showed...

We thank the reviewer for his/her remark and modified the text accordingly.

Page 16: The human body is composed of more than 600 different skeletal muscles, each with specific functions and properties.

Is redundant with Page 12: At least 640 different skeletal muscles can be identified in the human body, each with a specific form, architecture, position, and function.

We modified the text.

The authors should be more straightforward in their conclusions, and not only write that "their results suggest".

We have modified a few sentences "these results suggest" to be more straightforward.

Page 9 : Dos Santos should be ref 10.

We modified accordingly.

Page 24 and figure legends: data are presented as mean values +/- SEM, whereas in the figures it is only + SEM.

We modified accordingly.

The significance of the sequences provided in Fig. S7B, S8B and C and S9B and D should be clarified.

We better explain the significance of the sequences presented, they are in fact the junctions at the Cas9 recombination sites to show that no deletion or insertion has been created at the recombination site.

The reference of alpha bungarotoxin in table S1 is not an antibody reference.

We modified accordingly.

The sequence of the primers in Table S2 should be homogenized.

We apologize for the previous version and now homogenized the presented sequences.

Reviewer #3 (Remarks to the Author):

This manuscript reports the discovery of a super enhancer that controls expression of myosin genes in skeletal muscle. They use snATAC-seq and ChIP-seq to identify a large region of open chromatin in the myosin clusters, then generate mouse models to assess sufficiency and necessity of the super enhancer. Overall, the results are important and interesting for the field as it begins to tease apart the mechanisms by which myosin genes are regulated to control muscle fiber type and function. The majority of the claims are supported by the data but there are some instances of unclear interpretations, which should be adjusted prior to publication. The questions related to technical issues or controls are mainly minor. Specific comments are below.

We thank the reviewer of his/her enthusiasm and for his/her constructive remarks.

1. Genotyping for the Enh⁻ and Enh⁺ should be shown, especially related to the comment that 'Enh⁺ integrated 2 complete copies'. How many copies does the Enh⁻ line contain?

We now present how qPCR experiments were performed for the genotyping, the sequence of the primers used and the copy number in Enh⁻(1) and Enh⁺(2) lines.

2. The meaning for the results from the experiments deleting two independent cis-regulatory module within the super enhancer are not clear. The authors say that there are distinct enhancer elements that possess distinct functions and then show a schematic at the end of the figure. Do the data support the idea that Myh8 and Myh13 are down since the data are not significant? How is there is a reduction of Myh1 protein in 4D but no change in mRNA levels in 4E? The schematic at the end is also confusing. Specifically, there is no connection between enhancer B and Myh4 even though Myh4 is increased. Is there a way to simplify this model?

We agree with the Reviewer, we simplified the model that is now presented in Figure 8, as suggested also by Reviewer 1. We also modified the text and better explain the consequences of mutant A and B.

3. I am not sure if the data in Fig. 5 definitively show that the myosin gene promoters 'compete' for the super enhancer. Some manipulations result in deletions and thus fewer promoters, whereas others are inversions, so availability of promoters and spacing has been altered. Maybe 3D chromatin looping experiments would show competition.

According also to Reviewer 1 we toned down our conclusions. This reviewer is right that 4C in mutants could provide confirmation of promoter competition, but that is an amount of work that goes beyond what is feasible.

4. The arbitrary classification of muscles into three categories is not clear. I think the issue is that the logic for this is not explained.

We better explained the classification of muscles in three categories that is based on the phenotype of corresponding muscles in mutant A and in Myh(1-4)inv3'.

5. The idea that there are differences in regulation based on location within the muscle in one of the mutants (Myh2 upregulated in deep regions) could use more data for support. Were multiple planes of section analyzed to know if the Myh2 is regional in terms of the length of the fiber? Some quantification for this across animals would also be helpful?

We thank the reviewer for the suggestion and now added a supplementary Figure showing the continuous regionalization of the phenotype along the fiber (now SupFig10).

6. It is mentioned in the abstract that the work could explain how some regions of muscles or certain muscles are spared or more impacted in muscle diseases. This is a bit of an over-interpretation as there is not any presented that would directly support such an argument. Alternatively, perhaps the authors could more clearly explain their reasoning for the interpretation.

We thank the reviewer for his comment and have modified the text accordingly by explaining that adult skeletal muscles are constructed during embryogenesis at distinct anatomical positions and by distinct genetic cascades. This genetic heterogeneity has been suggested to participate in the susceptibility of specific muscle groups to different neuromuscular diseases, as referenced in our manuscript. By introducing perturbations within the fMyh locus, we also provide evidence that within each muscle, one group of myofibers can adapt its phenotype in a manner distinct from that adopted by a group of neighboring myofibers, suggesting an additional degree of heterogeneity.

Reviewers' Comments:

Reviewer #1:

Remarks to the Author:

The authors have successfully address most of my previous concerns. However, I am still not fully convinced but their answer to the results obtained using Myh(1-4)Del/Del mice and the two different Myh(1-4) inversion mouse models. Firstly, in Fig 5 the expression of Myl8 and Myl13 are still presented using different scales for the different mice, which somehow obscures the rather dramatic differences between the deletion and the two inversions. Why are Myl8 and Myl13 displaying much higher expression in the deletion model even when Myl1 and/or Myl4 are also inactive in the inversion models?. I am not claiming that promoter competition is not important, but at least in the context of the presented models, distance could be having an even larger effect. In the absence of additional mouse models, which I reckon can be very time consuming, the authors should explicitly comment on the differences in Myl8 and Myl13 expression between deletion and inversion models and offer plausible explanations.

Reviewer #3:

None

Reviewer #4:

Remarks to the Author:

The authors have dealt with my concerns.

REVIEWER COMMENTS.

Reviewer #1 (Remarks to the Author):

The authors have successfully address most of my previous concerns. However, I am still not fully convinced but their answer to the results obtained using Myh(1-4)Del/Del mice and the two different Myh(1-4) inversion mouse models. Firstly, in Fig 5 the expression of Myl8 and Myl13 are still presented using different scales for the different mice, which somehow obscures the rather dramatic differences between the deletion and the two inversions. Why are Myl8 and Myl13 displaying much higher expression in the deletion model even when Myl1 and/or Myl4 are also inactive in the inversion models?. I am not claiming that promoter competition is not important, but at least in the context of the presented models, distance could be having an even larger effect. In the absence of additional mouse models, which I reckon can be very time consuming, the authors should explicitly comment on the differences in Myl8 and Myl13 expression between deletion and inversion models and offer plausible explanations.

In the new Figure 5 we have changed the histograms presenting the relative amounts of *Myh* mRNAs, as requested by the reviewer, using the same scales. We agree with the reviewer that with this representation it is easier to see that the very large increase in *Myh8* expression observed in *Myh(1-4)Del* mice muscles is much larger than that observed in the *Myh(1-4)Inv* and *Myh(1-4)Inv3'* models.

In the *Myh(1-4)Del* mice expression of *Myh8* is strongly upregulated, *Myh8* promoter being further from the SE than *Myh2*. We interpret this result as the absence of the switch that normally occurs in the postnatal period between *Myh8* and *Myh4*: *Myh8* remains active, the transcription factors present in the fiber being more favorable to a SE-*Myh8* interaction than SE-*Myh2* in most myofibers.

In the case of *Myh(1-4)Inv* mice, where the *Myh1* and *Myh4* genes have been flipped, *Myh4* is found closer to the SE, nevertheless its activity is not increased, while *Myh1* expression is very strongly decreased. In this configuration the SE-*Myh8* distance remains unchanged. The decrease in *Myh1* expression is probably due, as we write, to the fact that not all *Myh1* regulatory regions have been inverted. SE-*Myh1* interactions are therefore reduced and transcription very low (Sup9J). This decrease would participate in the maintenance of SE-*Myh8* interactions in specific myofibers, with competition with *Myh1* strongly reduced. In this case, *Myh4* promoter is active (contrary to its inactivity in the *Myh(1-4)Del* model): we did not observe that it was inactive (Figure 5B, F, Sup 9 I), contrary to the sentence of the reviewer: "*Why are Myl8 and Myl13 displaying much higher expression in the deletion model even when Myl1 and/or Myl4 are also inactive in the inversion models?*". In the *Myh(1-4)Inv* model *Myh4* promoter is still active, and as it is the main promoter activated in limb muscles it remains an efficient competitor to avoid SE-*Myh8* or SE-*Myh13* interactions. In the *Myh(1-4)Inv* model MYH4 is still detected in the majority of limb myofibers (Figure 5B).

In *Myh(1-4)Inv3'* mice, *Myh4* transcription is not significantly different from that observed in *Myh(1-4)Inv* mice (Sup 9 I), but in *Myh(1-4)Inv3'* mice *Myh4* mRNAs are destabilized because they lack the last 3' exons; we reported that the 3' exons of *Myh4* were absent, and the truncated MYH4 protein is not detected. So the steady state of *Myh4* mRNA is strongly reduced (Figure 5G) but transcription still occurs. As the distance between SE and *Myh8* in wt and in the *Myh(1-4)Inv* mice is conserved, it is

decreased by only few kb in *Myh(1-4)Inv3'* mice. In *Myh(1-4)Inv3'*, *Myh4* promoter is active, and we did not observe that it was inactive, contrary to the sentence of the reviewer: "*Why are Myl8 and Myl13 displaying much higher expression in the deletion model even when Myl1 and/or Myl4 are also inactive in the inversion models?*". In the *Myh(1-4)Inv3'* model *Myh4* promoter is still active, and as it is the main promoter activated in limb muscles it remains an efficient competitor to avoid SE-*Myh8* or SE-*Myh13* interactions.

We discuss now the effect of distance on SE properties. We indicate that although it is farther from the SE the *Myh8* promoter is most active during development, compared with the closer *Myh2*, *Myh1* and *Myh4* promoters. During activation of the adult promoters, *Myh2*, *Myh1* and *Myh4*, *Myh4* which is furthest from the SE is activated in a majority of the leg myofibers. *Myh2* which is closer to the SE is activated mainly in the innermost regions of muscle masses.

We propose in the new discussion the possibility that the distance between the SE and the different *Myh* promoters could have a role and indicate that indeed the analysis of other mouse lines in particular where *Myh8* would be closer to the SE would allow to measure the importance of the distance between the SE and *Myh8* in the decrease of its activation in favor of *Myh4* during postnatal development. Nevertheless, we also suggest that the change in accumulation of specific transcription factors during postnatal development is responsible for the activation of *Myh4* and the other adult promoters *Myh2* and *Myh1*, at the expense of *Myh8*. We further discuss the factor distance between promoters and the SE in models of *Myh1/Myh4* deletion, or *Myh1/Myh4* inversion to explain the observed activation of *Myh8* and *Myh13*, keeping as a major idea the transcription factories present in the nuclei of muscle fibers programmed to execute a *Myh4* program.

The discussion has been expanded on page 17 (in red) to include the effect that the distance between the SE and the promoters of the locus might have on their activity. We thank the reviewer for raising this important aspect of locus regulation.

Reviewers' Comments:

Reviewer #1:

Remarks to the Author:

The authors have successfully address all my previous concerns and I would like to personally congratulate them for their very thorough and interesting work.

Also, I would like to apologize for not being able to revise this version of the manuscript earlier.